# Geometric Median (GM) Matching for Robust Data Pruning

## Abstract

Data pruning, the combinatorial task of selecting a small and informative subset from a large dataset, is crucial for mitigating the enormous computational costs associated with training data-hungry modern deep learning models at scale. Since large-scale data collections are invariably noisy, developing data pruning strategies that remain robust even in the presence of corruption is critical in practice. In response, we propose GM Matching – a herding (Welling, 2009) style greedy algorithm – that *yields a $k$-subset such that the mean of the subset approximates the geometric median of the (potentially) noisy dataset*. Theoretically, we show that GM Matching enjoys an improved $\mathcal{O}(1/k)$ scaling over $\mathcal{O}(1/\sqrt{k})$ scaling of uniform sampling; while achieving the optimal breakdown point of 1/2 even under arbitrary corruption. Extensive experiments across popular deep learning benchmarks indicate that GM Matching consistently outperforms prior state-of-the-art; the gains become more profound at high rates of corruption and aggressive pruning rates; making it a strong baseline for robust data pruning.

## 1 Introduction

Recent success of deep learning has been largely fueled by training gigantic models over vast amounts of training data (Radford et al., 2021; 2018; Brown et al., 2020; Kaplan et al., 2020; Hestness et al., 2017). Such large scale training, however is associated with enormous computational costs hindering the path to democratizing AI (Paul et al., 2021). Data pruning, the combinatorial task of downsizing a large training set into a small informative subset (Feldman, 2020; Agarwal et al., 2005; Muthukrishnan et al., 2005; Har-Peled, 2011; Feldman & Langberg, 2011), is a promising approach for reducing the enormous computational and storage costs of modern deep learning.

### Existing Data Pruning Strategies

Consequently, a large body of recent works have been proposed to solve the data selection problem. At a high level, there are two main directions: One set of data pruning approaches rely on some carefully designed **pruning metrics**, rank the training samples based on the scores and retain a fraction of them as representative samples (super samples), used for training the downstream model. For example, (Xia et al., 2022; Joshi & Mirzasoleiman, 2023; Sorscher et al., 2022) calculate the importance score of a sample in terms of the distance from the centroid of its corresponding class marginal. Samples closer to the centroid are considered most prototypical (easy) and those far from the centroid are treated as least prototypical (hard). A second set of works reformulate this problem as minimizing a **moment matching** objective (Chen et al., 2010; Campbell & Broderick, 2018; Dwivedi & Mackey, 2021) that aims to select a subset whose mean closely matches that of the entire dataset.

While this work primarily focuses on spatial approaches, it is worth mentioning that the canonical importance scoring criterion have been proposed in terms gradient norm (Paul et al., 2021; Needell et al., 2014), uncertainty (Pleiss et al., 2020) and forgetfulness (Toneva et al., 2018). Typically, samples closer to the class centroid in feature space tend to have lower gradient norms, exhibit lower uncertainty, and are harder to forget during training. In contrast, samples farther from the centroid generally have higher gradient norms, greater uncertainty, and are easier to forget (Paul et al., 2021; Sorscher et al., 2022; Xia et al., 2022). Moreover, (Mirzasoleiman et al., 2020) extended the moment-matching approach to the gradient space, selecting subsets that preserve the overall gradient statistics of the full dataset.

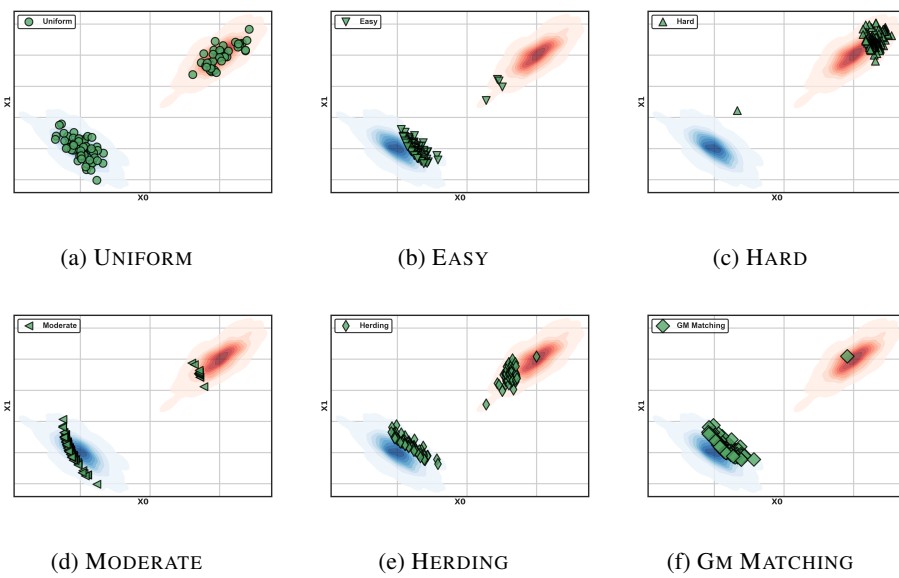

(a) UNIFORM        (b) EASY        (c) HARD

(d) MODERATE        (e) HERDING        (f) GM MATCHING

Figure 1: **DATA PRUNING IN THE WILD**: Data Pruning methods applied to samples from a multivariate Gaussian distribution (blue), with 40% replaced by an adversarial distribution (red). We subset 10% of the examples using: (UNIFORM) Random Sampling, (EASY) Selection of samples closest to the centroid. (HARD) Selection of samples farthest from the centroid. (MODERATE) Selection of samples closest to the median distance from the centroid. (HERDING) Moment Matching, (GM MATCHING) Robust Moment (GM) Matching (6). GM MATCHING yields significantly more robust (from the true distribution) subset than the other approaches.

## ROBUSTNESS VS DIVERSITY

In the **ideal scenario** (i.e. in absence of any corruption), hard examples are known to contribute the most in downstream generalization performance (Katharopoulos & Fleuret, 2018; Joshi et al., 2009; Huang et al., 2010; Balcan et al., 2007) as they often capture most of the usable information in the dataset (Xu et al., 2020). On the other hand, in **realistic noisy scenarios** involving outliers, this strategy often fails since the noisy examples are wrongly deemed informative for training (Zhang & Sabuncu, 2018; Park et al., 2024). Pruning methods specifically designed for such noisy scenarios thus propose to retain the most representative (easy) samples (Pleiss et al., 2020; Jiang et al., 2018; Har-Peled et al., 2006; Shah et al., 2020; Shen & Sanghavi, 2019). However, by only choosing samples far from the decision boundary, these methods ignore the more informative uncorrupted less prototypical samples. This can often result in sub-optimal downstream performance and in fact can also lead to degenerate solutions due to a covariance-shift problem (Sugiyama & Kawanabe, 2012); giving rise to a *robustness vs diversity trade off* (Xia et al., 2022). This restricts the applicability of existing pruning methods, as realistic scenarios often deviate from expected conditions, making it challenging or impractical to adjust the criteria and methods accordingly.

---

**Algorithm 1 GEOMETRIC MEDIAN MATCHING**

---

**Initialize :** A finite collection of $\alpha$ corrupted (Definition 1) observations $\mathcal{D}$ defined over Hilbert space $\mathcal{H} \in \mathbb{R}^d$, equipped with norm $\| \cdot \|$ and inner $\langle \cdot \rangle$ operators; initial weight vector $\boldsymbol{\theta}_0 \in \mathcal{H}$.

**Robust Mean Estimation:** $\boldsymbol{\mu}^{\text{GM}} = \arg \min_{\mathbf{z} \in \mathcal{H}} \sum_{\mathbf{x}_i \in \mathcal{D}} \|\mathbf{z} - \mathbf{x}_i\|$

$\mathcal{D}_{\mathcal{S}} \leftarrow \emptyset$

**for** *iterations t = 0, 1, …, k-1* **do**

     $\mathbf{x}_{t+1} := \arg \max_{\mathbf{x} \in \mathcal{D}} \langle \boldsymbol{\theta}_t, \mathbf{x} \rangle$

     $\boldsymbol{\theta}_{t+1} := \boldsymbol{\theta}_t + \boldsymbol{\mu}_\epsilon^{\text{GM}} - \mathbf{x}_{t+1}$

     $\mathcal{D}_{\mathcal{S}} := \mathcal{D}_{\mathcal{S}} \cup \mathbf{x}_{t+1}$

     $\mathcal{D} := \mathcal{D} \setminus \mathbf{x}_{t+1}$

**end**

**return:** $\mathcal{D}_{\mathcal{S}}$

---

OVERVIEW OF OUR APPROACH

To go beyond these limitations, we study data pruning in presence of corruption. Specifically, we consider the $\alpha$ corruption framework (Definition 1), where $0 \leq \psi < \frac{1}{2}$ fraction of the samples are allowed be **arbitrarily perturbed**. This allowance for **arbitrary corruption** enables us to generalize many practical robustness scenarios; including **corrupt feature / label** and **adversarial attacks**.

We make a key observation that, traditional pruning methods typically use the empirical mean to calculate the centroid of the samples, which then guides the selection process based on how representative those samples are. However, the empirical mean is highly susceptible to outliers – in fact, it is possible to construct a single adversarial example to arbitrarily perturb the empirical mean. As a consequence, in the presence of arbitrary corruption, the conventional distinction between easy (robust) and hard samples breaks down, leading to the selection of subsets that are significantly compromised by corruption as illustrated in Figure 1, depicting sampling from a corrupted Gaussian.

In response, we propose a data pruning strategy that fosters balanced diversity, effectively navigating various regions of the distribution while avoiding distant, noisy points. Our key idea is to replace the target moment in the standard moment matching objective with a robust surrogate – Geometric Median (Weber et al., 1929; Weiszfeld, 1937) – a classical robust estimator of the mean. In particular, we optimize over finding a subset minimizes the discrepancy between the subset's mean and the GM (Definition 3) of the (potentially noisy) dataset using greedy herding (Welling, 2009) style update rule. We call our algorithm Geometric Median Matching as described in Algorithm 1.

CONTRIBUTIONS

Overall, our contributions can be summarized as follows:

- We systematically and formally investigate and extend data pruning in presence of corruption. In particular, we study data pruning under the gross corruption framework (Definition 1), where up to 1/2 fraction of the training examples are allowed to be arbitrarily corrupted. We note that, existing pruning heuristics (including the ones proposed for robust scenarios) break down under this strong corruption, due to empirical mean's vulnerability to corruption (Section 4, Figure 1).

- Motivated by this key observation, we exploit the robustness property of GM (Definition 3), to design a novel robust moment matching objective (6). It aims at finding a subset such that the mean of the subset approximates the GM of the noisy dataset. We minimize over this objective using greedy herding (Welling, 2009) style update rule. We call the resulting data pruning algorithm GM MATCHING and formally describe it in Algorithm 1.

- Leveraging classical robustness properties of GM, we show that, GM Matching converges to a bounded neighborhood of original underlying mean, at an impressive $\mathcal{O}(1/k)$ rate while being robust even when up to 1/2 of the samples are arbitrarily corrupted ( Theorem 1) .

- Extensive experiments over CIFAR 10/100, Tiny ImageNet, across feature corruption, label noise and adversarial attacks indicate the superiority of GM Matching over existing methods. We improve over prior work almost in all settings, the gains are especially more profound (often by more than 10%) in presence of corruption and at aggressive pruning rates; making GM Matching a strong baseline for future research in robust data pruning.

## 2  PROBLEM SETUP : ROBUST DATA PRUNING

Given a set of samples $\mathcal{D}$, the goal of data pruning is to select a subset of the most representative samples $\mathcal{D}_\mathcal{S} \subseteq \mathcal{D}$, that can approximate the underlying distribution well. Data pruning methods achieve this by first defining a *pruning criterion* e.g. based on distance, uncertainty, diversity; and then selecting a subset that best satisfies these criteria to represent the full dataset effectively. If such a subset (also referred to as coreset) can be found in a compute efficient manner, then training a parametric model on the subset, typically yields similar generalization performance as training on the entire dataset while resulting in significant speed up when $|\mathcal{D}_\mathcal{S}| \ll |\mathcal{D}|$. However, for machine learning systems deployed in the wild, $\mathcal{D}$ is often noisy and imperfect due to the difficulty and expense of obtaining perfect semantic annotations for large amounts of data, adversarial attacks or simply measurement noises.

**Definition 1** ($\alpha$-**corruption**). *Given a set of observations from the original distribution of interest, an adversary is allowed to **inspect all the samples** and **arbitrarily perturb** up to $\psi \in [0, \frac{1}{2})$ fraction of them. We refer to a set of samples $\mathcal{D} = \mathcal{D}_\mathcal{G} \cup \mathcal{D}_\mathcal{B}$ as $\alpha$-corrupted , $\alpha := |\mathcal{D}_\mathcal{B}|/|\mathcal{D}_\mathcal{G}| = \frac{\psi}{1-\psi} < 1$ and $\mathcal{D}_\mathcal{B}$, $\mathcal{D}_\mathcal{G}$ denote the sets of corrupt and clean samples respectively.*

To this end, this work studies data pruning under the $\alpha$-corruption framework (Definition 1), where a fraction $\psi \in [0, \frac{1}{2})$ of the samples can be **arbitrarily** corrupted – a strong corruption model (Diakonikolas et al., 2019; Acharya et al., 2022) that generalizes the popular **Huber Contamination Model** (Huber, 1992), as well as the notorious **Byzantine Corruption** (Lamport et al., 1982).

> Given an $\alpha$-corrupted set of observations $\mathcal{D} = \mathcal{D}_\mathcal{G} \cup \mathcal{D}_\mathcal{B}$, the goal of ROBUST DATA PRUNING is thus to judiciously select a subset $\mathcal{D}_\mathcal{S} \subseteq \mathcal{D}$ ; that *encapsulates the the underlying clean (uncorrupted) distribution induced by subset $\mathcal{D}_\mathcal{G}$ without any a-priori knowledge about the corrupted samples*.

We measure the robustness of data pruning algorithms via breakdown point analysis (Donoho & Huber, 1983) – a classic tool in robust optimization to assess the resilience of an estimator.

**Definition 2** (**Breakdown Point**). *The breakdown point of an estimator is defined as the smallest fraction of contaminated data that can cause the estimator to result in arbitrarily large errors.*

In the context of Definition 1, we say that an estimator achieves **optimal breakdown point 1/2** (Lopuhaa et al., 1991) if it remains robust in presence of $\alpha$-corruption $\forall\ \alpha < 1$.

## 3 WARM UP : MOMENT MATCHING

In the uncorrupted setting i.e. when $\mathcal{D}_\mathcal{B} = \emptyset$, a natural and widely used approach for data pruning is to formulate it as the following combinatorial MOMENT MATCHING objective:

$$\underset{\mathcal{D}_\mathcal{S} \subseteq \mathcal{D}, |\mathcal{D}_\mathcal{S}|=k}{\arg\min} \left\| \frac{1}{|\mathcal{D}|} \sum_{\mathbf{x}_i \in \mathcal{D}} \mathbf{x}_i - \frac{1}{k} \sum_{\mathbf{x}_i \in \mathcal{D}_\mathcal{S}} \mathbf{x}_i \right\|^2 \tag{1}$$

Observe that, (1) is an instance of the famous set function maximization problem – known to be NP hard via a reduction from $k$-set cover (Feige, 1998). Despite its intractability, (Mirzasoleiman et al., 2020) demonstrated a transformation into a submodular set cover problem, enabling efficient solution via greedy algorithms (Nemhauser et al., 1978; Wolsey, 1982). The greedy approach: also referred to as kernel herding (Welling, 2009; Welling & Chen, 2010) starts with a suitably chosen $\boldsymbol{\theta}_0 \in \mathbb{R}^d$; and iteratively adds samples via the following update rule:

$$\mathbf{x}_{t+1} := \underset{\mathbf{x} \in \mathcal{D}}{\arg\max}\ \langle \boldsymbol{\theta}_t, \mathbf{x} \rangle \tag{2}$$

$$\boldsymbol{\theta}_{t+1} := \boldsymbol{\theta}_t + \left( \frac{1}{|\mathcal{D}|} \sum_{\mathbf{x}_i \in \mathcal{D}} \mathbf{x}_i - \mathbf{x}_{t+1} \right) \tag{3}$$

It's worth noting that this algorithm is an infinite memory, deterministic process as at each iteration T, $\boldsymbol{\theta}_T$ encapsulates the entire sampling history: $\boldsymbol{\theta}_T = \boldsymbol{\theta}_0 + T\boldsymbol{\mu} - \sum_{t=1}^{T} \mathbf{x}_t$ where $\boldsymbol{\mu} = \frac{1}{|\mathcal{D}|} \sum_{\mathbf{x}_i \in \mathcal{D}} \mathbf{x}_i$. Conceptually, $\boldsymbol{\theta}_T$ represents the vector pointing towards under-sampled regions of the target distribution induced by $\mathcal{D}$ at iteration $T$. The algorithm's greedy selection strategy aligns each new sample with $\boldsymbol{\theta}$, effectively *herding* new points to fill the gaps left by earlier selections. Remarkably, (Chen et al., 2010) showed that this simple greedy update rule achieves an impressive $\mathcal{O}(1/k)$ convergence rate for (1), a quadratic improvement over random sampling where the error decreases at the rate $\mathcal{O}(1/\sqrt{k})$. The result holds if $\|\mathbf{x}\| \leq R\ \forall \mathbf{x} \in \mathcal{D}$ for some constant $R$ and as long as the target moment is in the relative interior of $\mathcal{C} = \text{conv}\{\mathbf{x}|\mathbf{x} \in \mathcal{D}\}$ (Proposition 1 (Chen et al., 2010)).

## 4 GEOMETRIC MEDIAN (GM) MATCHING

Despite its strong performance guarantees in the vanilla (uncorrupted) setting, we argue that the algorithm can result in arbitrarily poor solution in the noisy setting. The vulnerability can be

attributed to the estimation of target moment via empirical mean – notorious for its sensitivity to outliers. Consider a single adversarial sample: $\mathbf{x}^{\mathcal{B}} = |\mathcal{D}|\boldsymbol{\mu}^{\mathcal{B}} - \sum_{\mathbf{x} \in \mathcal{D} \setminus \mathbf{x}^{\mathcal{B}}} \mathbf{x}$, shifting the empirical mean to adversary chosen arbitrary target $\boldsymbol{\mu}^{\mathcal{B}}$. This implies that the empirical mean can't tolerate even a single grossly corrupted sample i.e. yields **lowest possible asymptotic breakdown point of 0**. As a consequence, optimizing over the moment matching objective (1) no longer guarantee convergence to the true underlying (uncorrupted) moment $\boldsymbol{\mu}^{\mathcal{G}} = \mathbb{E}_{\mathbf{x} \in \mathcal{D}_{\mathcal{G}}} \mathbf{x}$, instead the algorithm can be hijacked by a single bad sample, warping the solution towards an adversarial target.

> Motivated by this key observation, a natural idea to enable ROBUST MOMENT MATCHING is to replace the empirical mean in (1) with a robust surrogate estimator of the target moment and perform greedy herding updates to match the robust surrogate. Ideally, the robust estimate $\boldsymbol{\mu}$ should ensure that the estimation error $\Delta = \|\boldsymbol{\mu} - \boldsymbol{\mu}^{\mathcal{G}}\| \leq \delta$ remain bounded, even when the observations are $\alpha$-corrupted (Definition 1). Moreover, the estimate should reside inside the relative interior of $\mathcal{C}_{\mathcal{G}} = \text{conv}\{\mathbf{x} | \mathbf{x} \in \mathcal{D}_{\mathcal{G}}\}$ to ensure the linear convergence guarantee.

In the univariate setting, various robust mean estimators, such as the median and the trimmed mean, are known to achieve the optimal breakdown point 1/2. A common strategy to extend these methods to the multivariate setting is to perform univariate estimation independently along each dimension. However, in high dimensions, these estimates need not lie in the convex hull of the samples and are not orthogonal equivariant and can even become degenerate in the overparameterized settings ($n \ll d$) (Lopuhaa et al., 1991; Rousseeuw & Leroy, 2005). On the other hand, M-estimators are affine equivariant but have breakdown point at most $1/(d+1)$ (Donoho & Huber, 1983).

> **Definition 3** (**Geometric Median**). *Given a finite collection of observations $\{\mathbf{x}_1, \mathbf{x}_2, \ldots \mathbf{x}_n\}$ defined over Hilbert space $\mathcal{H} \in \mathbb{R}^d$, equipped with norm $\|\cdot\|$ and inner $\langle \cdot \rangle$ operators, the geometric median(or Fermat-Weber point) (Weber et al., 1929) is defined as:*
>
> $$\boldsymbol{\mu}^{\text{GM}} = \text{GM}(\{\mathbf{x}_1, \mathbf{x}_2, \ldots \mathbf{x}_n\}) = \underset{\mathbf{z} \in \mathcal{H}}{\arg\min} \left[ \rho(\mathbf{z}) := \sum_{i=1}^{n} \left\| \mathbf{z} - \mathbf{x}_i \right\| \right] \qquad (4)$$

In this context, Geometric Median (GM) ( Definition 3) – a well studied spatial estimator, known for several nice properties like **rotation and translation invariance** and **optimal breakdown point of 1/2 under gross corruption** (Minsker et al., 2015; Kemperman, 1987). Moreover, the estimate is guaranteed to lie in the relative interior of the convex hull of the majority (good) points i.e. $\boldsymbol{\mu}^{\text{GM}} \in \mathcal{C}_{\mathcal{G}}$ making it a natural choice to estimate the target moment.

Computing the GM exactly, is known to be hard as linear time algorithm exists (Bajaj, 1988), making it is necessary to rely on approximation methods to estimate the geometric median (Weiszfeld, 1937; Vardi & Zhang, 2000; Cohen et al., 2016). We call a point $\boldsymbol{\mu}_{\epsilon}^{\text{GM}} \in \mathcal{H}$ an $\epsilon$ accurate GM if it holds:

$$\sum_{i=1}^{n} \left\| \boldsymbol{\mu}_{\epsilon}^{\text{GM}} - \mathbf{x}_i \right\| \leq (1 + \epsilon) \sum_{i=1}^{n} \left\| \boldsymbol{\mu}^{\text{GM}} - \mathbf{x}_i \right\| \qquad (5)$$

We then, exploit the breakdown and translation invariance property of GM and solve for the following ROBUST MOMENT MATCHING objective – a robust surrogate of (1):

$$\underset{\mathcal{D}_{\mathcal{S}} \subseteq \mathcal{D}, |\mathcal{D}_{\mathcal{S}}| = k}{\arg\min} \left\| \boldsymbol{\mu}_{\epsilon}^{\text{GM}} - \frac{1}{k} \sum_{\mathbf{x}_i \in \mathcal{S}} \mathbf{x}_i \right\|^2 \qquad (6)$$

Consequently, we perform herding style greedy minimization of the error (6) :

> We start with a suitably chosen $\boldsymbol{\theta}_0 \in \mathbb{R}^d$; and repeatedly perform the following updates, adding one sample at a time, $k$ times:
>
> $$\mathbf{x}_{t+1} := \underset{\mathbf{x} \in \mathcal{D}}{\arg\max} \langle \boldsymbol{\theta}_t, \mathbf{x} \rangle \qquad (7)$$
>
> $$\boldsymbol{\theta}_{t+1} := \boldsymbol{\theta}_t + \left( \boldsymbol{\mu}_{\epsilon}^{\text{GM}} - \mathbf{x}_{t+1} \right) \qquad (8)$$

We refer to the resulting robust data pruning approach as GM MATCHING. For ease of exposition, let $\boldsymbol{\theta}_0 = \boldsymbol{\mu}_\epsilon^{\text{GM}}$. Then, at iteration $t = T$, GM MATCHING is performing:

$$\mathbf{x}_{T+1} = \arg\max_{\mathbf{x} \in \mathcal{D}} \left[ \langle \boldsymbol{\mu}_\epsilon^{\text{GM}}, \mathbf{x} \rangle - \frac{1}{T+1} \sum_{t=1}^{T} \langle \mathbf{x}, \mathbf{x}_t \rangle \right] \tag{9}$$

Greedy updates in the direction that reduces the accumulated error, encourages the algorithm to explore underrepresented regions of the feature space, **promoting diversity**. By matching the GM rather than the empirical mean, the algorithm imposes larger penalties on outliers, which lie farther from the core distribution. This encourages GM MATCHING to **prioritize samples near the convex hull of uncorrupted points** $\mathcal{C}_\mathcal{G} = \text{conv}\{\phi_\mathcal{B}(\mathbf{x}) | \mathbf{x} \in \mathcal{D}_\mathcal{G}\}$. As a result, the algorithm promotes diversity in a balanced manner, effectively exploring different regions of the distribution while avoiding distant, noisy points, thus mitigating the robustness vs. diversity trade-off discussed in Section 1. This makes GM MATCHING an excellent choice for data pruning in the wild.

THEORETICAL GUARANTEE

In order to theoretically characterize the convergence behavior of GM MATCHING, we first exploit the robustness property of GM (Acharya et al., 2022; Cohen et al., 2016; Chen et al., 2017) to get an upper bound on the estimation error w.r.t the underlying true mean. Next, we use the property that GM is guaranteed to lie in the interior of the convex hull of majority of the samples (Minsker et al., 2015; Boyd & Vandenberghe, 2004) which follows from the properties of convex sets. Combining these two results we establish the following convergence guarantee for GM MATCHING :

---

**Theorem 1.** *Suppose that, we are given, a set of $\alpha$-corrupted samples $\mathcal{D} = \mathcal{D}_\mathcal{G} \cup \mathcal{D}_\mathcal{B}$ (Definition 1) and an $\epsilon$ approx. $\text{GM}(\cdot)$ oracle (4). Further assume that $\|\mathbf{x}\| \leq R \ \forall \mathbf{x} \in \mathcal{D}$ for some constant $R$. Then, GM MATCHING guarantees that the mean of the selected $k$-subset $\mathcal{D}_\mathcal{S} \subseteq \mathcal{D}$ converges to a $\delta$-neighborhood of the uncorrupted (true) mean $\boldsymbol{\mu}^\mathcal{G} = \mathbb{E}_{\mathbf{x} \in \mathcal{D}_\mathcal{G}}(\mathbf{x})$ at the rate $\overline{\mathcal{O}}(\frac{1}{k})$ such that:*

$$\delta^2 = \mathbb{E}\left\| \frac{1}{k} \sum_{\mathbf{x}_i \in \mathcal{D}_\mathcal{S}} \mathbf{x}_i - \boldsymbol{\mu}^\mathcal{G} \right\|^2 \leq \frac{8|\mathcal{D}_\mathcal{G}|}{(|\mathcal{D}_\mathcal{G}| - |\mathcal{D}_\mathcal{B}|)^2} \sum_{\mathbf{x} \in \mathcal{D}_\mathcal{G}} \mathbb{E}\left\| \mathbf{x} - \boldsymbol{\mu}^\mathcal{G} \right\|^2 + \frac{2\epsilon^2}{(|\mathcal{D}_\mathcal{G}| - |\mathcal{D}_\mathcal{B}|)^2} \tag{10}$$

---

This result suggest that, even in presence of $\alpha$ corruption, the proposed algorithm GM Matching converges to a neighborhood of the true mean, where the neighborhood radius depends on two terms – the first term depends on the variance of the uncorrupted samples and the second term depends on how accurately the GM is calculated. Furthermore the bound holds $\forall \alpha = \mathcal{D}_\mathcal{B}/\mathcal{D}_\mathcal{G} < 1$ implying GM Matching remains robust even when half of the samples are arbitrarily corrupted i.e. it achieves the optimal breakdown point of 1/2. The detailed proofs are provided in Section 8.

## 5 EXPERIMENTS

In this section, we outline our experimental setup, present our key empirical findings, and discuss deeper insights into the performance of GM Matching. Due to space constraint we only present a subset of the results in the main paper. Please refer to Section 8, for additional experimental evidence.

**BASELINES:** To ensure reproducibility, our experimental setup is identical to (Xia et al., 2022). We compare the proposed GM Matching selection strategy against the following popular data pruning strategies as baselines for comparison: (1) Random; (2) Herding Welling (2009); (3) Forgetting Toneva et al. (2018); (4) GraNd-score Paul et al. (2021); (5) EL2N-score Paul et al. (2021); (6) Optimization-based Yang et al. (2022); (7) Self-sup.-selection Sorscher et al. (2022) and (8) Moderate (Xia et al., 2022). We do not run these baselines for be these baselines are borrowed from (Xia et al., 2020). Additionally, for further ablations we compare GM Matching with many (natural) distance based geometric pruning strategies: (**UNIFORM**) Random Sampling, (**EASY**) Selection of samples closest to the centroid; (**HARD**) Selection of samples farthest from the centroid; (**MODERATE**) (Xia et al., 2022) Selection of samples closest to the median distance from the centroid; (**HERDING**) Moment Matching (Chen et al., 2010), (**GM MATCHING**) Robust Moment (GM) Matching (6).

**DATASETS AND NETWORKS:** We perform extensive experiments across three popular image classification datasets - CIFAR10, CIFAR100 and Tiny-ImageNet. Our experiments span popular

| | CIFAR-100 | | | | | | |
|---|---|---|---|---|---|---|---|
| Method / Ratio | 20% | 30% | 40% | 60% | 80% | 100% | Mean ↑ |
| Random | 50.26±3.24 | 53.61±2.73 | 64.32±1.77 | 71.03±0.75 | 74.12±0.56 | 78.14±0.55 | 62.67 |
| Herding | 48.39±1.42 | 50.89±0.97 | 62.99±0.61 | 70.61±0.44 | 74.21±0.49 | 78.14±0.55 | 61.42 |
| Forgetting | 35.57±1.40 | 49.83±0.91 | 59.65±2.50 | **73.34±0.39** | **77.50±0.53** | 78.14±0.55 | 59.18 |
| GraNd-score | 42.65±1.39 | 53.14±1.28 | 60.52±0.79 | 69.70±0.68 | 74.67±0.79 | 78.14±0.55 | 60.14 |
| EL2N-score | 27.32±1.16 | 41.98±0.54 | 50.47±1.20 | 69.23±1.00 | 75.96±0.88 | 78.14±0.55 | 52.99 |
| Optimization-based | 42.16±3.30 | 53.19±2.14 | 58.93±0.98 | 68.93±0.70 | 75.62±0.33 | 78.14±0.55 | 59.77 |
| Self-sup.-selection | 44.45±2.51 | 54.63±2.10 | 62.91±1.20 | 70.70±0.82 | 75.29±0.45 | 78.14±0.55 | 61.60 |
| Moderate-DS | 51.83±0.52 | 57.79±1.61 | 64.92±0.93 | 71.87±0.91 | 75.44±0.40 | 78.14±0.55 | 64.37 |
| **GM Matching** | **55.93± 0.48** | **63.08± 0.57** | **66.59± 1.18** | 70.82± 0.59 | 74.63± 0.86 | 78.14± 0.55 | **66.01** |
| | Tiny ImageNet | | | | | | |
| Random | 24.02±0.41 | 29.79±0.27 | 34.41±0.46 | 40.96±0.47 | 45.74±0.61 | 49.36±0.25 | 34.98 |
| Herding | 24.09±0.45 | 29.39±0.53 | 34.13±0.37 | 40.86±0.61 | 45.45±0.33 | 49.36±0.25 | 34.78 |
| Forgetting | 22.37±0.71 | 28.67±0.54 | 33.64±0.32 | 41.14±0.43 | **46.77±0.31** | 49.36±0.25 | 34.52 |
| GraNd-score | 23.56±0.52 | 29.66±0.37 | 34.33±0.50 | 40.77±0.42 | 45.96±0.56 | 49.36±0.25 | 34.86 |
| EL2N-score | 19.74±0.26 | 26.58±0.40 | 31.93±0.28 | 39.12±0.46 | 45.32±0.27 | 49.36±0.25 | 32.54 |
| Optimization-based | 13.88±2.17 | 23.75±1.62 | 29.77±0.94 | 37.05±2.81 | 43.76±1.50 | 49.36±0.25 | 29.64 |
| Self-sup.-selection | 20.89±0.42 | 27.66±0.50 | 32.50±0.30 | 39.64±0.39 | 44.94±0.34 | 49.36±0.25 | 33.13 |
| Moderate-DS | 25.29±0.38 | 30.57±0.20 | 34.81±0.51 | 41.45±0.44 | 46.06±0.33 | 49.36±0.25 | 35.64 |
| **GM Matching** | **27.88±0.19** | **33.15±0.26** | **36.92±0.40** | **42.48±0.12** | 46.75±0.51 | 49.36±0.25 | **37.44** |

Table 1: **No Corruption :** Comparing (Test Accuracy) pruning algorithms on CIFAR-100 and Tiny-ImageNet in the uncorrupted setting. ResNet-50 is used both as proxy and for downstream classification.

| Method / Selection ratio | 20% | 30% | 40% | 60% | 80% | 100% | Mean ↑ |
|---|---|---|---|---|---|---|---|
| | CIFAR-100 with 20% corrupted images | | | | | | |
| Random | 40.99±1.46 | 50.38±1.39 | 57.24±0.65 | 65.21±1.31 | 71.74±0.28 | 74.92±0.88 | 57.11 |
| Herding | 44.42±0.46 | 53.57±0.31 | 60.72±1.78 | 69.09±1.73 | 73.08±0.98 | 74.92±0.88 | 60.18 |
| Forgetting | 26.39±0.17 | 40.78±2.02 | 49.95±2.31 | 65.71±1.12 | 73.67±1.12 | 74.92±0.88 | 51.30 |
| GraNd-score | 36.33±2.66 | 46.21±1.48 | 55.51±0.76 | 64.59±2.40 | 70.14±1.36 | 74.92±0.88 | 54.56 |
| EL2N-score | 21.64±2.03 | 23.78±1.66 | 35.71±1.17 | 56.32±0.86 | 69.66±0.43 | 74.92±0.88 | 41.42 |
| Optimization-based | 33.42±1.60 | 45.37±2.81 | 54.06±1.74 | 65.19±1.27 | 70.06±0.83 | 74.92±0.88 | 54.42 |
| Self-sup.-selection | 42.61±2.44 | 54.04±1.90 | 59.51±1.22 | 68.97±0.96 | 72.33±0.20 | 74.92±0.88 | 60.01 |
| Moderate-DS | 42.98±0.87 | 55.80±0.95 | 61.84±1.96 | 70.05±1.29 | 73.67±0.30 | 74.92±0.88 | 60.87 |
| **GM Matching** | **47.12±0.64** | **59.17±0.92** | **63.45±0.34** | **71.70±0.60** | **74.60±1.03** | 74.92±0.88 | **63.21** |
| | Tiny ImageNet with 20 % corrupted images | | | | | | |
| Random | 19.99±0.42 | 25.93±0.53 | 30.83±0.44 | 37.98±0.31 | 42.96±0.62 | 46.68±0.43 | 31.54 |
| Herding | 19.46±0.14 | 24.47±0.33 | 29.72±0.39 | 37.50±0.59 | 42.28±0.30 | 46.68±0.43 | 30.86 |
| Forgetting | 18.47±0.46 | 25.53±0.23 | 31.17±0.24 | 39.35±0.44 | 44.55±0.67 | 46.68±0.43 | 31.81 |
| GraNd-score | 20.07±0.49 | 26.68±0.40 | 31.25±0.40 | 38.21±0.49 | 42.84±0.72 | 46.68±0.43 | 30.53 |
| EL2N-score | 18.57±0.30 | 24.42±0.44 | 30.04±0.15 | 37.62±0.44 | 42.43±0.61 | 46.68±0.43 | 30.53 |
| Optimization-based | 13.71±0.26 | 23.33±1.84 | 29.15±2.84 | 36.12±1.86 | 42.94±0.52 | 46.88±0.43 | 29.06 |
| Self-sup.-selection | 20.22±0.23 | 26.90±0.50 | 31.93±0.49 | 39.74±0.52 | 44.27±0.10 | 46.68±0.43 | 32.61 |
| Moderate-DS | 23.27±0.33 | 29.06±0.36 | 33.48±0.11 | 40.07±0.36 | 44.73±0.39 | 46.68±0.43 | 34.12 |
| **GM Matching** | **27.19±0.92** | **31.70±0.78** | **35.14±0.19** | **42.04±0.31** | **45.12±0.28** | 46.68±0.43 | **36.24** |

Table 2: **Image Corruption :** Experiments comparing pruning methods when 20% of the images are corrupted. ResNet-50 is used for both proxy (data pruning) and downstream training.

deep nets including ResNet-18/50 (He et al., 2016), VGG-16 (Simonyan & Zisserman, 2014), ShuffleNet (Ma et al., 2018), SENet (Hu et al., 2018), EfficientNet-B0(Tan & Le, 2019).

**IMPLEMENTATION DETAILS:** For the CIFAR-10/100 experiments, we utilize a batch size of 128 and employ SGD optimizer with a momentum of 0.9, weight decay of 5e-4, and an initial learning rate of 0.1. The learning rate is reduced by a factor of 5 after the 60th, 120th, and 160th epochs, with a total of 200 epochs. Data augmentation techniques include random cropping and random horizontal flipping. In the Tiny-ImageNet experiments, a batch size of 256 is used with an SGD optimizer, momentum of 0.9, weight decay of 1e-4, and an initial learning rate of 0.1. The learning rate is decreased by a factor of 10 after the 30th and 60th epochs, with a total of 90 epochs. Random horizontal flips are applied for data augmentation. Each experiment is repeated over 5 random seeds and the variances are noted. Throughout this paper, we use Weiszfield Solver (Weiszfeld, 1937) to compute GM approximately.

| Method / Ratio | CIFAR-100 (Label noise) | | Tiny ImageNet (Label noise) | | Mean ↑ |
|---|---|---|---|---|---|
| | 20% | 30% | 20% | 30% | |
| **20% Label Noise** | | | | | |
| Random | 34.47±0.64 | 43.26±1.21 | 17.78±0.44 | 23.88±0.42 | 29.85 |
| Herding | 42.29±1.75 | 50.52±3.38 | 18.98±0.44 | 24.23±0.29 | 34.01 |
| Forgetting | 36.53±1.11 | 45.78±1.04 | 13.20±0.38 | 21.79±0.43 | 29.33 |
| GraNd-score | 31.72±0.67 | 42.80±0.30 | 18.28±0.32 | 23.72±0.18 | 28.05 |
| EL2N-score | 29.82±1.19 | 33.62±2.35 | 13.93±0.69 | 18.57±0.31 | 23.99 |
| Optimization-based | 32.79±0.62 | 41.80±1.14 | 14.77±0.95 | 22.52±0.77 | 27.57 |
| Self-sup.-selection | 31.08±0.78 | 41.87±0.63 | 15.10±0.73 | 21.01±0.36 | 27.27 |
| Moderate-DS | 40.25±0.12 | 48.53±1.60 | 19.64±0.40 | 24.96±0.30 | 31.33 |
| **GM Matching** | **52.64±0.72** | **61.01±0.47** | **25.80±0.37** | **31.71±0.24** | **42.79** |
| **35% Label Noise** | | | | | |
| Random | 24.51±1.34 | 32.26±0.81 | 14.64±0.29 | 19.41±0.45 | 22.71 |
| Herding | 29.42±1.54 | 37.50±2.12 | 15.14±0.45 | 20.19±0.45 | 25.56 |
| Forgetting | 29.48±1.98 | 38.01±2.21 | 11.25±0.90 | 17.07±0.66 | 23.14 |
| GraNd-score | 23.03±1.05 | 34.83±2.01 | 13.68±0.46 | 19.51±0.45 | 22.76 |
| EL2N-score | 21.95±1.08 | 31.63±2.84 | 10.11±0.25 | 13.69±0.32 | 19.39 |
| Optimization-based | 26.77±0.15 | 35.63±0.92 | 12.37±0.68 | 18.52±0.90 | 23.32 |
| Self-sup.-selection | 23.12±1.47 | 34.85±0.68 | 11.23±0.32 | 17.76±0.69 | 22.64 |
| Moderate-DS | 28.45±0.53 | 36.55±1.26 | 15.27±0.31 | 20.33±0.28 | 25.15 |
| **GM Matching** | **43.33± 1.02** | **58.41± 0.68** | **23.14± 0.92** | **27.76± 0.40** | **38.16** |

Table 3: **Robustness to Label Noise:** Comparing (Test Accuracy) pruning methods on CIFAR-100 and TinyImageNet datasets, under 20% and 35% Symmetric Label Corruption, at 20% and 30% selection ratio. ResNet-50 is used both as proxy and for downstream classification.

**PROXY MODEL:** Needless to say, identifying sample importance is an ill-posed problem without some notion of similarity among the samples. Thus, it is common to assume access to a proxy encoder that maps the features to a separable embedding space – a property often satisfied by off-the-shelf pretrained foundation models (Hessel et al., 2021; Sorscher et al., 2022). We perform experiments across multiple choices of such proxy encoder scenarios: **(A) Standard Setting:** when the proxy model shares the same architecture as the model Table 1- 4). Additionally, we also experiment with **(B) Distribution Shift:** proxy model pretrained on a different (distribution shifted) dataset( Figure 2-3) e.g. ImageNet and used to sample from CIFAR10. **(C) Network Transfer:** where, the proxy has a different network compared to the downstream classifier (Table 5).

IDEAL (NO CORRUPTION) SCENARIO

Our first sets of experiments involve performing data pruning across selection ratio ranging from 20% - 80% in the uncorrupted setting. The corresponding results, presented in Table 1, indicate that while GM Matching is developed with robustness scenarios in mind, it outperforms the existing strong baselines even in the clean setting. Overall, on both CIFAR-100 and Tiny ImageNet GM Matching improves over the prior methods > 2% on an average. In particular, we note that GM Matching enjoys larger gains in the low data selection regime, while staying competitive at low pruning rates.

CORRUPTION SCENARIOS

To understand the performance of data pruning strategies in presence of corruption, we experiment with three different sources of corruption – image corruption, label noise and adversarial attacks.

**ROBUSTNESS TO IMAGE CORRUPTION:** In this set of experiments, we investigate the robustness of data pruning strategies when the input images are corrupted – a popular robustness setting, often encountered when training models on real-world data (Hendrycks & Dietterich, 2019; Szegedy et al., 2013). To corrupt images, we apply five types of realistic noise: Gaussian noise, random occlusion, resolution reduction, fog, and motion blur to parts of the corrupt samples i.e. to say if $m$ samples are corrupted, each type of noise is added to one a random $m/5$ of them, while the other partitions are corrupted with a different noise. The results are presented in Table 2. We observe that GM Matching outperforms all the baselines across all pruning rates improving ≈3% across both datasets on an average. We note that, the gains are more consistent and profound in this setting over the clean setting. Additionally, similar to our prior observations in the clean setting, the gains of GM Matching are more significant at high pruning rates.

| | CIFAR-100 (PGD Attack) | | CIFAR-100 (GS Attack) | | |
|---|---|---|---|---|---|
| Method / Ratio | 20% | 30% | 20% | 30% | Mean ↑ |
| Random | 43.23±0.31 | 52.86±0.34 | 44.23±0.41 | 53.44±0.44 | 48.44 |
| Herding | 40.21±0.72 | 49.62±0.65 | 39.92±1.03 | 50.14±0.15 | 44.97 |
| Forgetting | 35.90±1.30 | 47.37±0.99 | 37.55±0.53 | 46.88±1.91 | 41.93 |
| GraNd-score | 40.87±0.84 | 50.13±0.30 | 40.77±1.11 | 49.88±0.83 | 45.41 |
| EL2N-score | 26.61±0.58 | 34.50±1.02 | 26.72±0.66 | 35.55±1.30 | 30.85 |
| Optimization-based | 38.29±1.77 | 46.25±1.82 | 41.36±0.92 | 49.10±0.81 | 43.75 |
| Self-sup.-selection | 40.53±1.15 | 49.95±0.50 | 40.74±1.66 | 51.23±0.25 | 45.61 |
| Moderate-DS | 43.60±0.97 | 51.66±0.39 | 44.69±0.68 | 53.71±0.37 | 48.42 |
| GM Matching | **45.41 ±0.86** | **51.80 ±1.01** | **49.78 ±0.27** | **55.50 ±0.31** | **50.62** |
| | Tiny ImageNet (PGD Attack) | | Tiny ImageNet (GS Attack) | | |
| Method / Ratio | 20% | 30% | 20% | 30% | Mean ↑ |
| Random | 20.93±0.30 | 26.60±0.98 | 22.43±0.31 | 26.89±0.31 | 24.21 |
| Herding | 21.61±0.36 | 25.95±0.19 | 23.04±0.28 | 27.39±0.14 | 24.50 |
| Forgetting | 20.38±0.47 | 26.12±0.19 | 22.06±0.31 | 27.21±0.21 | 23.94 |
| GraNd-score | 20.76±0.21 | 26.34±0.32 | 22.56±0.30 | 27.52±0.40 | 24.30 |
| EL2N-score | 16.67±0.62 | 22.36±0.42 | 19.93±0.57 | 24.65±0.32 | 20.93 |
| Optimization-based | 19.26±0.77 | 24.55±0.92 | 21.26±0.24 | 25.88±0.37 | 22.74 |
| Self-sup.-selection | 19.23±0.46 | 23.92±0.51 | 19.70±0.20 | 24.73±0.39 | 21.90 |
| Moderate-DS | 21.81±0.37 | 27.11±0.20 | 23.20±0.13 | 28.89±0.27 | 25.25 |
| GM Matching | **25.98 ±1.12** | **30.77 ±0.25** | **29.71 ±0.45** | **32.88 ±0.73** | **29.84** |

Table 4: **Robustness to Adversarial Attacks**. Comparing (Test Accuracy) pruning methods under PGD and GS attacks. ResNet-50 is used both as proxy and for downstream classification.

| | ResNet-50→SENet | | ResNet-50→EfficientNet-B0 | | |
|---|---|---|---|---|---|
| Method / Ratio | 20% | 30% | 20% | 30% | Mean ↑ |
| Random | 34.13±0.71 | 39.57±0.53 | 32.88±1.52 | 39.11±0.94 | 36.42 |
| Herding | 34.86±0.55 | 38.60±0.68 | 32.21±1.54 | 37.53±0.22 | 35.80 |
| Forgetting | 33.40±0.64 | 39.79±0.78 | 31.12±0.21 | 38.38±0.65 | 35.67 |
| GraNd-score | 35.12±0.54 | 41.14±0.42 | 33.20±0.67 | 40.02±0.35 | 37.37 |
| EL2N-score | 31.08±1.11 | 38.26±0.45 | 31.34±0.49 | 36.88±0.32 | 34.39 |
| Optimization-based | 33.18±0.52 | 39.42±0.77 | 32.16±0.90 | 38.52±0.50 | 35.82 |
| Self-sup.-selection | 31.74±0.71 | 38.45±0.39 | 30.99±1.03 | 37.96±0.77 | 34.79 |
| Moderate-DS | 36.04±0.15 | 41.40±0.20 | 34.26±0.48 | 39.57±0.29 | 37.82 |
| GM Matching | **37.93±0.23** | **42.59±0.29** | **36.31±0.67** | **41.03±0.41** | **39.47** |

Table 5: **Network Transfer (Clean)** : Tiny-ImageNet Model Transfer Results. A ResNet-50 proxy is used to find important samples which are then used to train SENet and EfficientNet.

**ROBUSTNESS TO LABEL CORRUPTION:** Next, we consider another important corruption scenario where a fraction of the training examples are mislabeled. We conduct experiments with synthetically injected symmetric label noise (Li et al., 2022; Patrini et al., 2017; Xia et al., 2020). The results are summarized in Table 3. Encouragingly, GM Matching **outperforms the baselines by ≈ 12%**. Since, mislabeled samples come from different class - they tend to be spatially quite dissimilar, being less likely to be picked by GM matching, explaining the superior performance.

**ROBUSTNESS TO ADVERSARIAL ATTACKS:** Finally, we experiment with adversarial attacks that add imperceptible but adversarial noise on natural examples (Szegedy et al., 2013; Huang et al., 2010). Specifically, we employ two popular adversarial attack algorithms – PGD attack (Madry et al., 2017) and GS Attacks (Goodfellow et al., 2014) on models trained with CIFAR-100 and Tiny-ImageNet to generate adversarial examples. Following this, various pruning methods are applied to these adversarial examples, and the models are retrained on the curated subset of data. The results are summarized in Table 4. Similar to other corruption scenarios, even in this setting, GM MATCHING outperforms the baselines yielding ≈ 3% average gain over the best performing baseline.

GENERALIZATION TO UNSEEN NETWORK / DOMAIN

Since, the input features (e.g. images) often reside on a non-separable manifold, data pruning strategies rely on a proxy model to map the samples into a separable manifold (embedding space), wherein the data pruning strategies can now assign importance scores. However, it is important for the data pruning strategies to be robust to architecture changes i.e. to say that samples selected via a

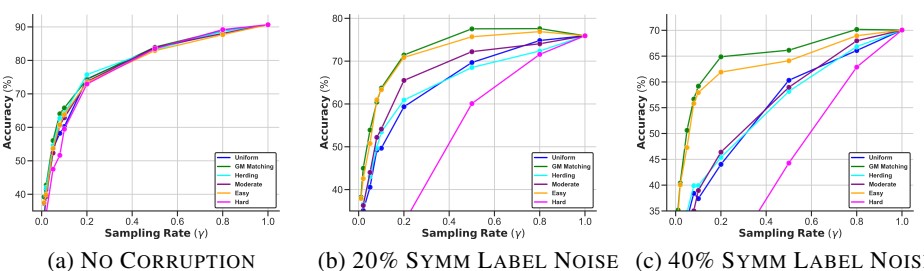

(a) NO CORRUPTION     (b) 20% SYMM LABEL NOISE     (c) 40% SYMM LABEL NOISE

Figure 2: **Domain Transfer ( ImageNet-1k → CIFAR-10 ) Proxy :** CIFAR10, corrupted with label noise is pruned using a (proxy) ResNet-18 pretrained on ImageNet-1k. A ResNet-18 is trained from scratch on the subset. We compare our method GM MATCHING with geometric pruning baselines: UNIFORM, EASY, HARD, MODERATE, HERDING.

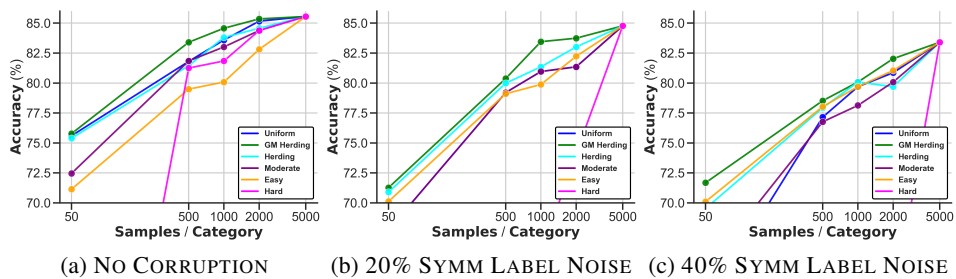

(a) NO CORRUPTION     (b) 20% SYMM LABEL NOISE     (c) 40% SYMM LABEL NOISE

Figure 3: **Domain Transfer ( ImageNet-1k → CIFAR-10 ) Proxy + Embedding :** We train a Linear Classifier on CIFAR10; over embeddings obtained from a frozen ResNet-18 pretrained on ImageNet-1k. The dataset was pruned using the same encoder. We compare our method GM MATCHING with geometric pruning baselines: UNIFORM, EASY, HARD, MODERATE, HERDING across different label noise settings.

proxy network should generalize well when trained on unseen (during sample selection) networks / domains. We perform experiments on two such scenarios:

**NETWORK TRANSFER:** In this setting, the proxy model is trained on the target dataset (no distribution shift). However, the proxy architecture is different than the downstream network. In Table 5, we use a ResNet-50 proxy trained on Mini-ImageNet to sample the data. However, then we train a downstream SENet and EfficientNet-B0 on the sampled data.

**DOMAIN TRANSFER:** Next, we consider the setting where the proxy shares the same architecture with the downstream model. However, the proxy used to select the samples is pretrained on a different dataset (distribution shift) than target dataset. In Figure 2 we use a proxy ResNet-18 pretrained on ImageNet to select samples from CIFAR10. The selected samples are used to train a subsequent ResNet-18. In Figure 3, we additionally freeze the pretrained encoder i.e. we use ResNet-18 encoder pretrained on ImageNet as proxy. Further, we freeze the encoder and train a downstream linear classifier on top over CIFAR-10.

## 6 CONCLUSION

In this work, we formalized the problem of robust data pruning. We show that existing data pruning strategies suffer significant degradation in performance in presence of corruption. Orthogonal to existing works, we propose GM MATCHING where our goal is to find a $k$-subset from the noisy data such that the mean of the subset approximates the GM of the noisy dataset. We solve this meta problem using a herding style greedy approach. We theoretically justify our approach and empirically show its efficacy by comparing it against several popular benchmarks across multiple datasets. Our results indicate that GM MATCHING consistently outperforms existing pruning strategies in both clean and noisy settings making it a lucrative tool for data pruning in the wild.

## 7 REPRODUCIBILITY STATEMENT

We provide the source code implementation of the proposed algorithm as well as a notebook with a running demo on Synthetic Gaussian Dataset. The hyper-parameters and other training details to reproduce our benchmarks are provided in Section 5. Several benchmarks for existing methods were borrowed directly from prior work, in such cases the source has been appropriately cited e.g. (Xia et al., 2022). All the proofs have been stated clearly in Appendix with necessary assumptions.

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

# Supplementary Material for GM MATCHING

CONTENTS:

# 8 APPENDIX

## 8.1 NOTATIONS AND ABBREVIATIONS

| | |
|---|---|
| $a$ | A scalar (integer or real) |
| $\mathbf{a}$ | A vector |
| $\mathbf{A}$ | A matrix |
| $\mathrm{a}$ | A scalar random variable |
| $\mathbf{a}$ | A vector-valued random variable |
| $\mathbb{A}, \mathcal{A}$ | A set |
| $[a, b]$ | The real interval including $a$ and $b$ |
| $\mathbb{A} \backslash \mathbb{B}$ | Set subtraction, i.e., the set containing the elements of $\mathbb{A}$ that are not in $\mathbb{B}$ |
| $\mathbf{a}_i$ | Element $i$ of the random vector $\mathbf{a}$ |
| $P(\mathrm{a})$ | A probability distribution over a discrete variable |
| $p(\mathrm{a})$ | A probability distribution over a continuous variable, or over a variable whose type has not been specified |
| $f : \mathbb{A} \to \mathbb{B}$ | The function $f$ with domain $\mathbb{A}$ and range $\mathbb{B}$ |
| $f \circ g$ | Composition of the functions $f$ and $g$ |
| $f(\mathbf{x}; \boldsymbol{\theta})$ | A function of $\mathbf{x}$ parametrized by $\boldsymbol{\theta}$. (Sometimes we write $f(\mathbf{x})$ and omit the argument $\boldsymbol{\theta}$ to lighten notation) |
| $\lvert\lvert\mathbf{x}\rvert\rvert_p$ | $L^p$ norm of $\mathbf{x}$ |
| $\mathbf{1}(condition)$ | is 1 if the condition is true, 0 otherwise |
| GM MATCHING | Geometric Median Matching |

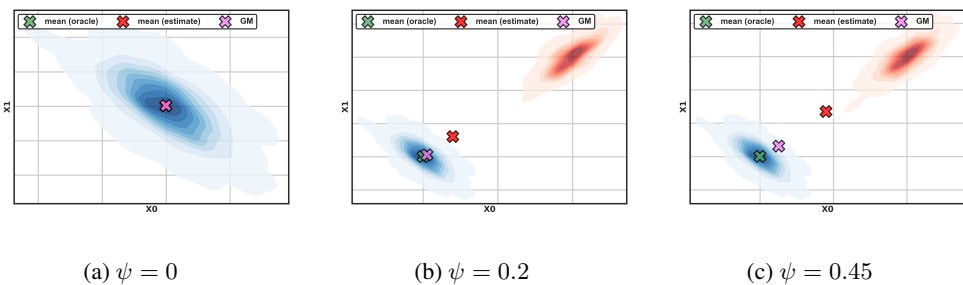

(a) $\psi = 0$       (b) $\psi = 0.2$       (c) $\psi = 0.45$

Figure 4: **ROBUST MEAN ESTIMATION**: As we progressively increase $0 \leq \psi < 1/2$ (fraction of corrupt samples in the data); while the empirical mean drifts away, GM remains close to the uncorrupted mean.

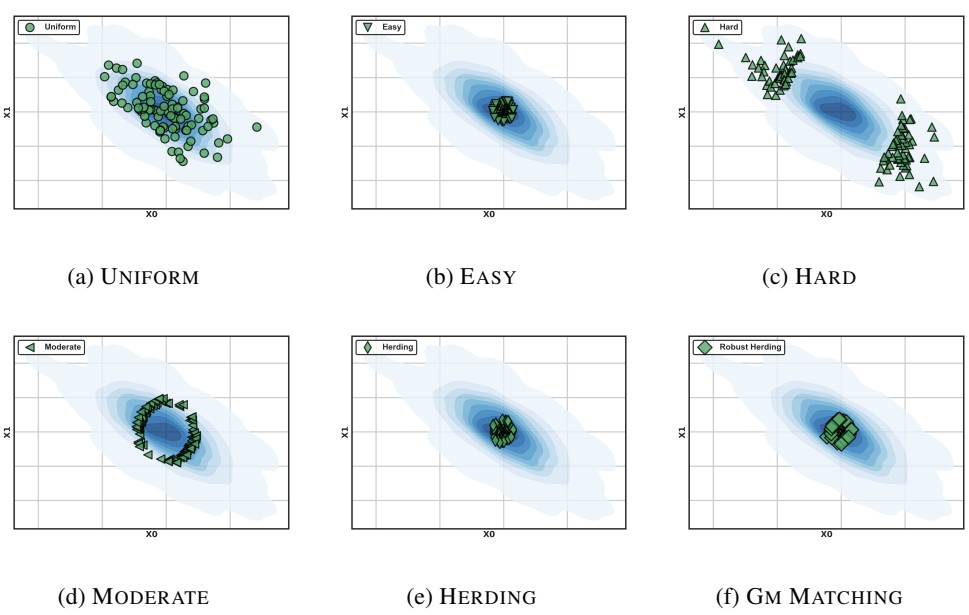

(a) UNIFORM       (b) EASY       (c) HARD

(d) MODERATE       (e) HERDING       (f) GM MATCHING

Figure 5: **No Corruption :** We select 10% of the samples using: (UNIFORM) Random Sampling, (EASY) Selection of samples closest to the centroid. (HARD) Selection of samples farthest from the centroid. (MODERATE) Selection of samples closest to the median distance from the centroid. (HERDING) Moment Matching, (GM MATCHING) Robust Moment (GM) Matching (6).

## 8.2 TOY EXPERIMENTS

We simulate a Gaussian Mixture Model (GMM) with clean and adversarial components to evaluate robust moment estimation in noisy datasets. The clean data, is drawn from a Gaussian distribution with mean $[0, 0]$ and covariance $\begin{bmatrix} 1 & 0.5 \\ 0.5 & 1 \end{bmatrix}$, while the adversarial data, is drawn from a Gaussian with mean $[-5, 5]$ and the same covariance. We generate 1000 samples, forming a corrupted dataset by combining the clean and adversarial data.

- In Figure 4, we compare the mean of the corrupted dataset (noisy moment) with a robustly estimated mean using the geometric median to mitigate adversarial influence.
- Additionally, in Figure 5-7, we compare different geometric pruning strategies in the toy setting.

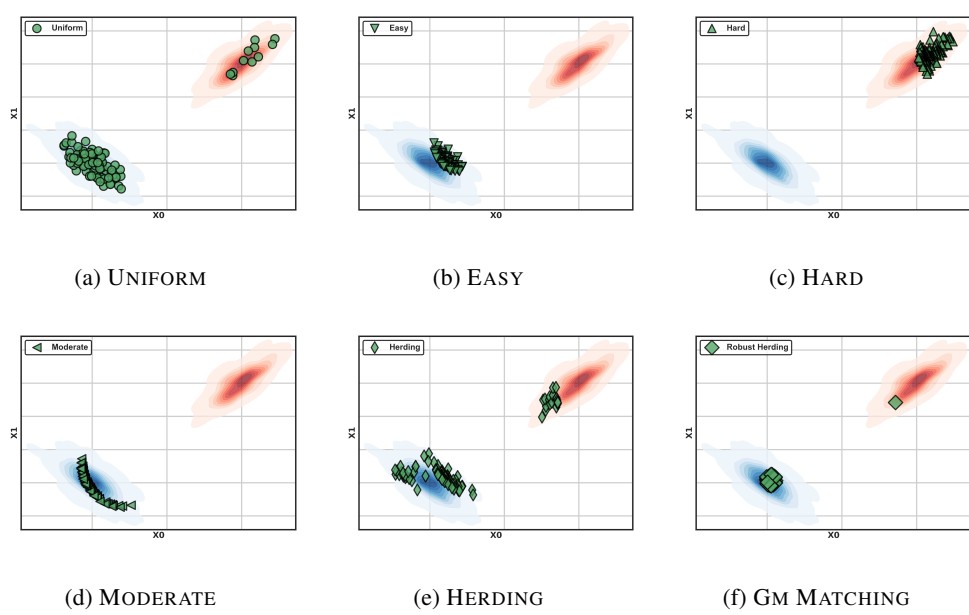

| (a) UNIFORM | (b) EASY | (c) HARD |
| (d) MODERATE | (e) HERDING | (f) GM MATCHING |

Figure 6: **20% Corruption**: In this experiment, 20% of the samples are corrupted – drawn from a adversary chosen distribution (red). We select 10% samples using: (UNIFORM) Random Sampling, (EASY) Selection of samples closest to the centroid. (HARD) Selection of samples farthest from the centroid. (MODERATE) Selection of samples closest to the median distance from the centroid. (HERDING) Moment Matching, (GM MATCHING) Robust Moment (GM) Matching (6). We see that while EASY remains robust, it is clearly sampling from low-density areas – failing to capture the prototypical samples.

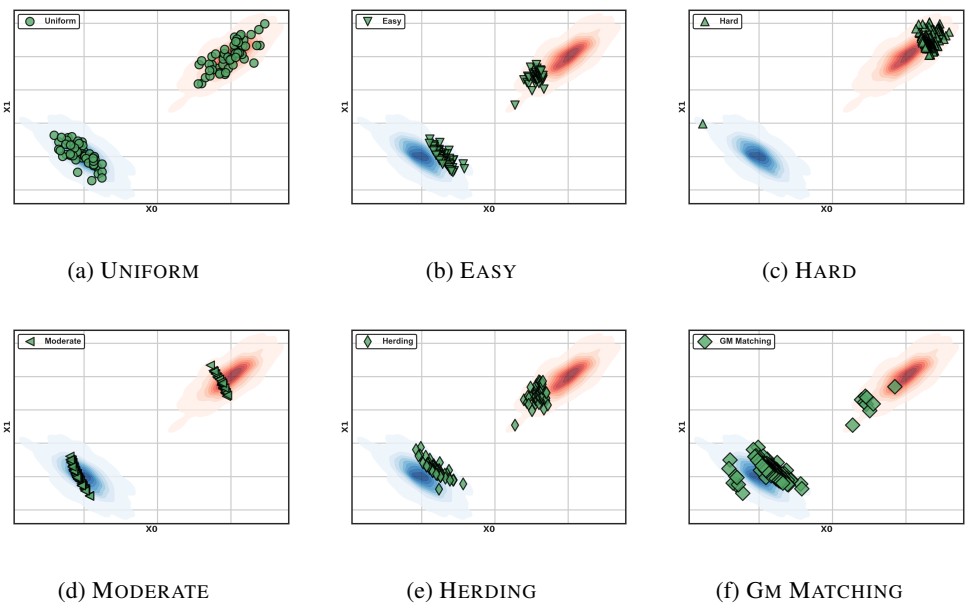

| (a) UNIFORM | (b) EASY | (c) HARD |
| (d) MODERATE | (e) HERDING | (f) GM MATCHING |

Figure 7: **Toy Example: 45% of the samples are corrupted** i.e. drawn from an adversary chosen distribution (red). We compare several baselines for choosing 10% samples: (UNIFORM) random sampling, (EASY) selects of samples closest to the centroid. (HARD) Selection of samples farthest from the centroid. (MODERATE) selects samples closest to the median distance from the centroid. (HERDING) moment matching, (GM MATCHING) robust moment (GM) matching (6). Clearly GM Matching is significantly more robust and diverse than the other approaches even at such high corruption rates.

| Method / Selection ratio | 20% | 30% | 40% | 60% | 80% | 100% | Mean ↑ |
|---|---|---|---|---|---|---|---|
| **CIFAR-100** | | | | | | | |
| **No Corruption** | | | | | | | |
| Random | 50.26±3.24 | 53.61±2.73 | 64.32±1.77 | 71.03±0.75 | 74.12±0.56 | 78.14±0.55 | 62.67 |
| Herding | 48.39±1.42 | 50.89±0.97 | 62.99±0.61 | 70.61±0.44 | 74.21±0.49 | 78.14±0.55 | 61.42 |
| Forgetting | 35.57±1.40 | 49.83±0.91 | 59.65±2.50 | **73.34±0.39** | **77.50±0.53** | 78.14±0.55 | 59.18 |
| GraNd-score | 42.65±1.39 | 53.14±1.28 | 60.52±0.79 | 69.70±0.68 | 74.67±0.79 | 78.14±0.55 | 60.14 |
| EL2N-score | 27.32±1.16 | 41.98±0.54 | 50.47±1.20 | 69.23±1.00 | 75.96±0.88 | 78.14±0.55 | 52.99 |
| Optimization-based | 42.16±3.30 | 53.19±2.14 | 58.93±0.98 | 68.93±0.70 | 75.62±0.33 | 78.14±0.55 | 59.77 |
| Self-sup.-selection | 44.45±2.51 | 54.63±2.10 | 62.91±1.20 | 70.70±0.82 | 75.29±0.45 | 78.14±0.55 | 61.60 |
| Moderate-DS | 51.83±0.52 | 57.79±1.61 | 64.92±0.93 | 71.87±0.91 | 75.44±0.40 | 78.14±0.55 | 64.37 |
| **GM Matching** | **55.93± 0.48** | **63.08± 0.57** | **66.59± 1.18** | 70.82± 0.59 | 74.63± 0.86 | 78.14± 0.55 | **66.01** |
| **5% Feature Corruption** | | | | | | | |
| Random | 43.14±3.04 | 54.19±2.92 | 64.21±2.39 | 69.50±1.06 | 72.90±0.52 | 77.26±0.39 | 60.79 |
| Herding | 42.50±1.27 | 53.88±3.07 | 60.54±0.94 | 69.15±0.55 | 73.47±0.89 | 77.26±0.39 | 59.81 |
| Forgetting | 32.42±0.74 | 49.72±1.64 | 54.84±2.20 | 70.22±2.00 | 75.19±0.40 | 77.26±0.39 | 56.48 |
| GraNd-score | 42.24±0.57 | 53.48±0.76 | 60.17±1.66 | 69.16±0.81 | 73.35±0.81 | 77.26±0.39 | 59.68 |
| EL2N-score | 26.13±1.16 | 39.01±1.42 | 49.89±1.87 | 68.36±1.41 | 73.10±0.36 | 77.26±0.39 | 51.30 |
| Optimization-based | 38.25±3.04 | 50.88±6.07 | 57.26±0.93 | 68.02±0.39 | 73.77±0.56 | 77.26±0.39 | 57.64 |
| Self-sup.-selection | 44.24±0.48 | 55.99±1.21 | 61.03±0.59 | 69.96±1.07 | 74.56±1.17 | 77.26±0.39 | 61.16 |
| Moderate-DS | 46.78±1.90 | 57.36±1.22 | 65.40±1.19 | 71.46±0.19 | **75.64±0.61** | 77.26±0.39 | 63.33 |
| **GM Matching** | **49.50±0.72** | **60.23±0.88** | **66.25±0.51** | **72.91±0.26** | 75.10±0.29 | 77.26±0.39 | **64.80** |
| **10% Feature Corruption** | | | | | | | |
| Random | 43.27±3.01 | 53.94±2.78 | 62.17±1.29 | 68.41±1.21 | 73.50±0.73 | 76.50±0.63 | 60.26 |
| Herding | 44.34±1.07 | 53.31±1.49 | 60.13±0.38 | 68.20±0.74 | 74.34±1.07 | 76.50±0.63 | 60.06 |
| Forgetting | 30.43±0.70 | 47.50±1.43 | 53.16±0.44 | 70.36±0.82 | 75.10±0.71 | 76.50±0.63 | 55.31 |
| GraNd-score | 36.36±1.06 | 52.26±0.66 | 60.22±1.39 | 68.96±0.62 | 72.78±0.51 | 76.50±0.63 | 58.12 |
| EL2N-score | 21.75±1.56 | 30.80±2.23 | 41.06±1.23 | 64.82±1.48 | 73.47±1.30 | 76.50±0.63 | 46.38 |
| Optimization-based | 37.22±0.39 | 48.92±1.38 | 56.88±1.48 | 67.33±2.15 | 72.94±1.90 | 76.50±0.63 | 56.68 |
| Self-sup.-selection | 42.01±1.31 | 54.47±1.19 | 61.37±0.68 | 68.52±1.24 | 74.73±0.36 | 76.50±0.63 | 60.22 |
| Moderate-DS | 47.02±0.66 | 55.60±1.67 | 62.18±1.86 | 71.83±0.78 | **75.66±0.66** | 76.50±0.63 | 62.46 |
| **GM Matching** | **48.86±1.02** | **60.15±0.43** | **66.92±0.28** | **72.03±0.38** | 73.71±0.19 | 76.50±0.63 | **64.33** |
| **20% Feature Corruption** | | | | | | | |
| Random | 40.99±1.46 | 50.38±1.39 | 57.24±0.65 | 65.21±1.31 | 71.74±0.28 | 74.92±0.88 | 57.11 |
| Herding | 44.42±0.46 | 53.57±0.31 | 60.72±1.78 | 69.09±1.73 | 73.08±0.98 | 74.92±0.88 | 60.18 |
| Forgetting | 26.39±0.17 | 40.78±2.02 | 49.95±2.31 | 65.71±1.12 | 73.67±1.12 | 74.92±0.88 | 51.30 |
| GraNd-score | 36.33±2.66 | 46.21±1.48 | 55.51±0.76 | 64.59±2.40 | 70.14±1.36 | 74.92±0.88 | 54.56 |
| EL2N-score | 21.64±2.03 | 23.78±1.66 | 35.71±1.17 | 56.32±0.86 | 69.66±0.43 | 74.92±0.88 | 41.42 |
| Optimization-based | 33.42±1.60 | 45.37±2.81 | 54.06±1.74 | 65.19±1.27 | 70.06±0.83 | 74.92±0.88 | 54.42 |
| Self-sup.-selection | 42.61±2.44 | 54.04±1.90 | 59.51±1.22 | 68.97±0.96 | 72.33±0.20 | 74.92±0.88 | 60.01 |
| Moderate-DS | 42.98±0.87 | 55.80±0.95 | 61.84±1.96 | 70.05±1.29 | 73.67±0.30 | 74.92±0.88 | 60.87 |
| **GM Matching** | **47.12±0.64** | **59.17±0.92** | **63.45±0.34** | **71.70±0.60** | **74.60±1.03** | 74.92±0.88 | **63.21** |

Table 6: **Image Corruption ( CIFAR 100 ):** Comparing (Test Accuracy) pruning methods when 20% of the images are corrupted. ResNet-50 is used both as proxy and for downstream classification.

## 8.3 ADDITIONAL BENCHMARK EXPERIMENTS

We share additional results on benchmark datasets that was omitted from the main paper due to space constraint. Table 6-12.

| Tiny ImageNet | | | | | | |
|---|---|---|---|---|---|---|
| **Method / Ratio** | **20%** | **30%** | **40%** | **60%** | **80%** | **100%** | **Mean ↑** |
| **No Corruption** | | | | | | |
| Random | 24.02±0.41 | 29.79±0.27 | 34.41±0.46 | 40.96±0.47 | 45.74±0.61 | 49.36±0.25 | 34.98 |
| Herding | 24.09±0.45 | 29.39±0.53 | 34.13±0.37 | 40.86±0.61 | 45.45±0.33 | 49.36±0.25 | 34.78 |
| Forgetting | 22.37±0.71 | 28.67±0.54 | 33.64±0.32 | 41.14±0.43 | **46.77±0.31** | 49.36±0.25 | 34.52 |
| GraNd-score | 23.56±0.52 | 29.66±0.37 | 34.33±0.50 | 40.77±0.42 | 45.96±0.56 | 49.36±0.25 | 34.86 |
| EL2N-score | 19.74±0.26 | 26.58±0.40 | 31.93±0.28 | 39.12±0.46 | 45.32±0.27 | 49.36±0.25 | 32.54 |
| Optimization-based | 13.88±2.17 | 23.75±1.62 | 29.77±0.94 | 37.05±2.81 | 43.76±1.50 | 49.36±0.25 | 29.64 |
| Self-sup.-selection | 20.89±0.42 | 27.66±0.50 | 32.50±0.30 | 39.64±0.39 | 44.94±0.34 | 49.36±0.25 | 33.13 |
| Moderate-DS | 25.29±0.38 | 30.57±0.20 | 34.81±0.51 | 41.45±0.44 | 46.06±0.33 | 49.36±0.25 | 35.64 |
| **GM Matching** | **27.88±0.19** | **33.15±0.26** | **36.92±0.40** | **42.48±0.12** | 46.75±0.51 | 49.36±0.25 | **37.44** |
| **5% Feature Corruption** | | | | | | |
| Random | 23.51±0.22 | 28.82±0.72 | 32.61±0.68 | 39.77±0.35 | 44.37±0.34 | 49.02±0.35 | 33.82 |
| Herding | 23.09±0.53 | 28.67±0.37 | 33.09±0.32 | 39.71±0.31 | 45.04±0.15 | 49.02±0.35 | 33.92 |
| Forgetting | 21.36±0.28 | 27.72±0.43 | 33.45±0.21 | 40.92±0.45 | 45.99±0.51 | 49.02±0.35 | 33.89 |
| GraNd-score | 22.47±0.23 | 28.85±0.83 | 33.81±0.24 | 40.40±0.15 | 44.86±0.49 | 49.02±0.35 | 34.08 |
| EL2N-score | 18.98±0.72 | 25.96±0.28 | 31.07±0.63 | 38.65±0.36 | 44.21±0.68 | 49.02±0.35 | 31.77 |
| Optimization-based | 13.65±1.26 | 24.02±1.35 | 29.65±1.86 | 36.55±1.84 | 43.64±0.71 | 49.02±0.35 | 29.50 |
| Self-sup.-selection | 19.35±0.57 | 26.11±0.31 | 31.90±0.37 | 38.91±0.29 | 44.43±0.42 | 49.02±0.35 | 32.14 |
| Moderate-DS | 24.63±0.78 | 30.27±0.16 | 34.84±0.24 | 40.86±0.42 | 45.60±0.31 | 49.02±0.35 | 35.24 |
| **GM Matching** | **27.46±1.22** | **33.14±0.61** | **35.76±1.14** | **41.62±0.71** | **46.83±0.56** | 49.02±0.35 | **36.96** |
| **10% Feature Corruption** | | | | | | |
| Random | 22.67±0.27 | 28.67±0.52 | 31.88±0.30 | 38.63±0.36 | 43.46±0.20 | 48.40±0.32 | 33.06 |
| Herding | 22.01±0.18 | 27.82±0.11 | 31.82±0.26 | 39.37±0.18 | 44.18±0.27 | 48.40±0.32 | 33.04 |
| Forgetting | 20.06±0.48 | 27.17±0.36 | 32.31±0.22 | 40.19±0.29 | 45.51±0.48 | 48.40±0.32 | 33.05 |
| GraNd-score | 21.52±0.48 | 26.98±0.43 | 32.70±0.19 | 40.03±0.26 | 44.87±0.35 | 48.40±0.32 | 33.22 |
| EL2N-score | 18.59±0.13 | 25.23±0.18 | 30.37±0.22 | 38.44±0.32 | 44.32±1.07 | 48.40±0.32 | 31.39 |
| Optimization-based | 14.05±1.74 | 29.18±1.77 | 29.12±0.61 | 36.28±1.88 | 43.52±0.31 | 48.40±0.32 | 29.03 |
| Self-sup.-selection | 19.47±0.26 | 26.51±0.55 | 31.78±0.14 | 38.87±0.54 | 44.69±0.29 | 48.40±0.32 | 32.26 |
| Moderate-DS | 23.79±0.16 | 29.56±0.16 | 34.60±0.12 | 40.36±0.27 | 45.10±0.23 | 48.40±0.32 | 34.68 |
| **GM Matching** | **27.41±0.23** | **32.84±0.98** | **36.27±0.68** | **41.85±0.29** | **46.35±0.44** | 48.40±0.32 | **36.94** |
| **20% Feature Corruption** | | | | | | |
| Random | 19.99±0.42 | 25.93±0.53 | 30.83±0.44 | 37.98±0.31 | 42.96±0.62 | 46.68±0.43 | 31.54 |
| Herding | 19.46±0.14 | 24.47±0.33 | 29.72±0.39 | 37.50±0.59 | 42.28±0.30 | 46.68±0.43 | 30.86 |
| Forgetting | 18.47±0.46 | 25.53±0.23 | 31.17±0.24 | 39.35±0.44 | 44.55±0.67 | 46.68±0.43 | 31.81 |
| GraNd-score | 20.07±0.49 | 26.68±0.40 | 31.25±0.40 | 38.21±0.49 | 42.84±0.72 | 46.68±0.43 | 30.53 |
| EL2N-score | 18.57±0.30 | 24.42±0.44 | 30.04±0.15 | 37.62±0.44 | 42.43±0.61 | 46.68±0.43 | 30.53 |
| Optimization-based | 13.71±0.26 | 23.33±1.84 | 29.15±2.84 | 36.12±1.86 | 42.94±0.52 | 46.88±0.43 | 29.06 |
| Self-sup.-selection | 20.22±0.23 | 26.90±0.50 | 31.93±0.49 | 39.74±0.52 | 44.27±0.10 | 46.68±0.43 | 32.61 |
| Moderate-DS | 23.27±0.33 | 29.06±0.36 | 33.48±0.11 | 40.07±0.36 | 44.73±0.39 | 46.68±0.43 | 34.12 |
| **GM Matching** | **27.19±0.92** | **31.70±0.78** | **35.14±0.19** | **42.04±0.31** | **45.12±0.28** | 46.68±0.43 | **36.24** |

Table 7: **Image Corruption ( Tiny ImageNet ):** Comparing (Test Accuracy) pruning methods under feature (image) corruption. ResNet-50 is used both as proxy and for downstream classification.

| Method / Ratio | CIFAR-100 (Label noise) | | Tiny ImageNet (Label noise) | | Mean ↑ |
|---|---|---|---|---|---|
| | 20% | 30% | 20% | 30% | |
| **20% Label Noise** | | | | | |
| Random | 34.47±0.64 | 43.26±1.21 | 17.78±0.44 | 23.88±0.42 | 29.85 |
| Herding | 42.29±1.75 | 50.52±3.38 | 18.98±0.44 | 24.23±0.29 | 34.01 |
| Forgetting | 36.53±1.11 | 45.78±1.04 | 13.20±0.38 | 21.79±0.43 | 29.33 |
| GraNd-score | 31.72±0.67 | 42.80±0.30 | 18.28±0.32 | 23.72±0.18 | 28.05 |
| EL2N-score | 29.82±1.19 | 33.62±2.35 | 13.93±0.69 | 18.57±0.31 | 23.99 |
| Optimization-based | 32.79±0.62 | 41.80±1.14 | 14.77±0.95 | 22.52±0.77 | 27.57 |
| Self-sup.-selection | 31.08±0.78 | 41.87±0.63 | 15.10±0.73 | 21.01±0.36 | 27.27 |
| Moderate-DS | 40.25±0.12 | 48.53±1.60 | 19.64±0.40 | 24.96±0.30 | 31.33 |
| **GM Matching** | **52.64±0.72** | **61.01±0.47** | **25.80±0.37** | **31.71±0.24** | **42.79** |
| **35% Label Noise** | | | | | |
| Random | 24.51±1.34 | 32.26±0.81 | 14.64±0.29 | 19.41±0.45 | 22.71 |
| Herding | 29.42±1.54 | 37.50±2.12 | 15.14±0.45 | 20.19±0.45 | 25.56 |
| Forgetting | 29.48±1.98 | 38.01±2.21 | 11.25±0.90 | 17.07±0.66 | 23.14 |
| GraNd-score | 23.03±1.05 | 34.83±2.01 | 13.68±0.46 | 19.51±0.45 | 22.76 |
| EL2N-score | 21.95±1.08 | 31.63±2.84 | 10.11±0.25 | 13.69±0.32 | 19.39 |
| Optimization-based | 26.77±0.15 | 35.63±0.92 | 12.37±0.68 | 18.52±0.90 | 23.32 |
| Self-sup.-selection | 23.12±1.47 | 34.85±0.68 | 11.23±0.32 | 17.76±0.69 | 22.64 |
| Moderate-DS | 28.45±0.53 | 36.55±1.26 | 15.27±0.31 | 20.33±0.28 | 25.15 |
| **GM Matching** | **43.33± 1.02** | **58.41± 0.68** | **23.14± 0.92** | **27.76± 0.40** | **38.16** |

Table 8: **Robustness to Label Noise:** Comparing (Test Accuracy) pruning methods on CIFAR-100 and TinyImageNet datasets, under 20% and 35% Symmetric Label Corruption, at 20% and 30% selection ratio. ResNet-50 is used both as proxy and for downstream classification.

| Method / Ratio | CIFAR-100 (Label noise) | | Tiny ImageNet (Label noise) | | Mean ↑ |
|---|---|---|---|---|---|
| | 20% | 30% | 20% | 30% | |
| Random | 24.51±1.34 | 32.26±0.81 | 14.64±0.29 | 19.41±0.45 | |
| Herding | 29.42±1.54 | 37.50±2.12 | 15.14±0.45 | 20.19±0.45 | |
| Forgetting | 29.48±1.98 | 38.01±2.21 | 11.25±0.90 | 17.07±0.66 | |
| GraNd-score | 23.03±1.05 | 34.83±2.01 | 13.68±0.46 | 19.51±0.45 | |
| EL2N-score | 21.95±1.08 | 31.63±2.84 | 10.11±0.25 | 13.69±0.32 | |
| Optimization-based | 26.77±0.15 | 35.63±0.92 | 12.37±0.68 | 18.52±0.90 | |
| Self-sup.-selection | 23.12±1.47 | 34.85±0.68 | 11.23±0.32 | 17.76±0.69 | |
| Moderate-DS | 28.45±0.53 | 36.55±1.26 | 15.27±0.31 | 20.33±0.28 | |
| **GM Matching** | **43.33± 1.02** | **58.41± 0.68** | **23.14± 0.92** | **27.76± 0.40** | |

Table 9: 35% Label Noise

| Method / Ratio | Tiny ImageNet (Label Noise) | | | | | | |
|---|---|---|---|---|---|---|---|
| | 20% | 30% | 40% | 60% | 80% | 100% | Mean ↑ |
| Random | 17.78±0.44 | 23.88±0.42 | 27.97±0.39 | 34.88±0.51 | 38.47±0.40 | 44.42±0.47 | 28.60 |
| Herding | 18.98±0.44 | 24.23±0.29 | 27.28±0.31 | 34.36±0.29 | 39.00±0.49 | 44.42±0.47 | 28.87 |
| Forgetting | 13.20±0.38 | 21.79±0.43 | 27.89±0.22 | **36.03±0.24** | **40.60±0.31** | 44.42±0.47 | 27.50 |
| GraNd-score | 18.28±0.32 | 23.72±0.18 | 27.34±0.33 | 34.91±0.19 | 39.45±0.45 | 44.42±0.47 | 28.34 |
| EL2N-score | 13.93±0.69 | 18.57±0.31 | 24.56±0.34 | 32.14±0.49 | 37.64±0.41 | 44.42±0.47 | 25.37 |
| Optimization-based | 14.77±0.95 | 22.52±0.77 | 25.62±0.90 | 34.18±0.79 | 38.49±0.69 | 44.42±0.47 | 27.12 |
| Self-sup.-selection | 15.10±0.73 | 21.01±0.36 | 26.62±0.22 | 33.93±0.36 | 39.22±0.12 | 44.42±0.47 | 27.18 |
| Moderate-DS | **19.64±0.40** | **24.96±0.30** | **29.56±0.21** | 35.79±0.36 | 39.93±0.23 | 44.42±0.47 | 30.18 |
| **GM Matching** | **25.80±0.37** | **31.71±0.24** | **34.87±0.21** | **39.76±0.71** | **41.94±0.23** | 44.42±0.47 | **34.82** |

Table 10: **Pruning with Label Noise (TinyImageNet):** Comparing (Test Accuracy) pruning methods under 20% Symmetric Label Corruption across wide array of selection ratio. ResNet-50 is used both as proxy and for downstream classification.

| | CIFAR-100 (PGD Attack) | | CIFAR-100 (GS Attack) | | |
|---|---|---|---|---|---|
| **Method / Ratio** | 20% | 30% | 20% | 30% | **Mean ↑** |
| Random | 43.23±0.31 | 52.86±0.34 | 44.23±0.41 | 53.44±0.44 | 48.44 |
| Herding | 40.21±0.72 | 49.62±0.65 | 39.92±1.03 | 50.14±0.15 | 44.97 |
| Forgetting | 35.90±1.30 | 47.37±0.99 | 37.55±0.53 | 46.88±1.91 | 41.93 |
| GraNd-score | 40.87±0.84 | 50.13±0.30 | 40.77±1.11 | 49.88±0.83 | 45.41 |
| EL2N-score | 26.61±0.58 | 34.50±1.02 | 26.72±0.66 | 35.55±1.30 | 30.85 |
| Optimization-based | 38.29±1.77 | 46.25±1.82 | 41.36±0.92 | 49.10±0.81 | 43.75 |
| Self-sup.-selection | 40.53±1.15 | 49.95±0.50 | 40.74±1.66 | 51.23±0.25 | 45.61 |
| Moderate-DS | 43.60±0.97 | 51.66±0.39 | 44.69±0.68 | 53.71±0.37 | 48.42 |
| **GM Matching** | **45.41 ±0.86** | **51.80 ±1.01** | **49.78 ±0.27** | **55.50 ±0.31** | **50.62** |
| | **Tiny ImageNet (PGD Attack)** | | **Tiny ImageNet (GS Attack)** | | |
| **Method / Ratio** | 20% | 30% | 20% | 30% | **Mean ↑** |
| Random | 20.93±0.30 | 26.60±0.98 | 22.43±0.31 | 26.89±0.31 | 24.21 |
| Herding | 21.61±0.36 | 25.95±0.19 | 23.04±0.28 | 27.39±0.14 | 24.50 |
| Forgetting | 20.38±0.47 | 26.12±0.19 | 22.06±0.31 | 27.21±0.21 | 23.94 |
| GraNd-score | 20.76±0.21 | 26.34±0.32 | 22.56±0.30 | 27.52±0.40 | 24.30 |
| EL2N-score | 16.67±0.62 | 22.36±0.42 | 19.93±0.57 | 24.65±0.32 | 20.93 |
| Optimization-based | 19.26±0.77 | 24.55±0.92 | 21.26±0.24 | 25.88±0.37 | 22.74 |
| Self-sup.-selection | 19.23±0.46 | 23.92±0.51 | 19.70±0.20 | 24.73±0.39 | 21.90 |
| Moderate-DS | 21.81±0.37 | 27.11±0.20 | 23.20±0.13 | 28.89±0.27 | 25.25 |
| **GM Matching** | **25.98 ±1.12** | **30.77 ±0.25** | **29.71 ±0.45** | **32.88 ±0.73** | **29.84** |

Table 11: **Robustness to Adversarial Attacks**. Comparing (Test Accuracy) pruning methods under PGD and GS attacks. ResNet-50 is used both as proxy and for downstream classification.

| Method / Ratio | ResNet-50→ VGG-16 | | ResNet-50→ ShuffleNet | | Mean ↑ |
|---|---|---|---|---|---|
| | 20% | 30% | 20% | 30% | |
| **No Corruption** | | | | | |
| Random | 29.63±0.43 | 35.38±0.83 | 32.40±1.06 | 39.13±0.81 | 34.96 |
| Herding | 31.05±0.22 | 36.27±0.57 | 33.10±0.39 | 38.65±0.22 | 35.06 |
| Forgetting | 27.53±0.36 | 35.61±0.39 | 27.82±0.56 | 36.26±0.51 | 32.35 |
| GraNd-score | 29.93±0.95 | 35.61±0.39 | 29.56±0.46 | 37.40±0.38 | 33.34 |
| EL2N-score | 26.47±0.31 | 33.19±0.51 | 28.18±0.27 | 35.81±0.29 | 31.13 |
| Optimization-based | 25.92±0.64 | 34.82±1.29 | 31.37±1.14 | 38.22±0.78 | 32.55 |
| Self-sup.-selection | 25.16±1.10 | 33.30±0.94 | 29.47±0.56 | 36.68±0.36 | 31.45 |
| Moderate-DS | 31.45±0.32 | 37.89±0.36 | 33.32±0.41 | 39.68±0.34 | 35.62 |
| **GM Matching** | **35.86±0.41** | **40.56±0.22** | **35.51±0.32** | **40.30±0.58** | **38.47** |
| **20% Label Corruption** | | | | | |
| Random | 23.29±1.12 | 28.18±1.84 | 25.08±1.32 | 31.44±1.21 | 27.00 |
| Herding | 23.99±0.36 | 28.57±0.40 | 26.25±0.47 | 30.73±0.28 | 27.39 |
| Forgetting | 14.52±0.66 | 21.75±0.23 | 15.70±0.29 | 22.31±0.35 | 18.57 |
| GraNd-score | 22.44±0.46 | 27.95±0.29 | 23.64±0.10 | 30.85±0.21 | 26.22 |
| EL2N-score | 15.15±1.25 | 23.36±0.30 | 18.01±0.44 | 24.68±0.34 | 20.30 |
| Optimization-based | 22.93±0.58 | 24.92±2.50 | 25.82±1.70 | 30.19±0.48 | 25.97 |
| Self-sup.-selection | 18.39±1.30 | 25.77±0.87 | 22.87±0.54 | 29.80±0.36 | 24.21 |
| Moderate-DS | 23.68±0.19 | 28.93±0.19 | 28.82±0.33 | 32.39±0.21 | 28.46 |
| **GM Matching** | **28.77±0.77** | **34.87±0.23** | **32.05±0.93** | **37.43±0.25** | **33.28** |
| **20% Feature Corruption** | | | | | |
| Random | 26.33±0.88 | 31.57±1.31 | 29.15±0.83 | 34.72±1.00 | 30.44 |
| Herding | 18.03±0.33 | 25.77±0.34 | 23.33±0.43 | 31.73±0.38 | 24.72 |
| Forgetting | 19.41±0.57 | 28.35±0.16 | 18.44±0.57 | 31.09±0.61 | 24.32 |
| GraNd-score | 23.59±0.19 | 30.69±0.13 | 23.15±0.56 | 31.58±0.95 | 27.25 |
| EL2N-score | 24.60±0.81 | 31.49±0.33 | 26.62±0.34 | 33.91±0.56 | 29.16 |
| Optimization-based | 25.12±0.34 | 30.52±0.89 | 28.87±1.25 | 34.08±1.92 | 29.65 |
| Self-sup.-selection | 26.33±0.21 | 33.23±0.26 | 26.48±0.37 | 33.54±0.46 | 29.90 |
| Moderate-DS | 29.65±0.68 | 35.89±0.53 | 32.30±0.38 | 38.66±0.29 | 34.13 |
| GM Matching | **33.45±1.02** | **39.46±0.44** | **35.14±0.21** | **39.89±0.98** | **36.99** |
| **PGD Attack** | | | | | |
| Random | 26.12±1.09 | 31.98±0.78 | 28.28±0.90 | 34.59±1.18 | 30.24 |
| Herding | 26.76±0.59 | 32.56±0.35 | 28.87±0.48 | 35.43±0.22 | 30.91 |
| Forgetting | 24.55±0.57 | 31.83±0.36 | 23.32±0.37 | 31.82±0.15 | 27.88 |
| GraNd-score | 25.19±0.33 | 31.46±0.54 | 26.03±0.66 | 33.22±0.24 | 28.98 |
| EL2N-score | 21.73±0.47 | 27.66±0.32 | 22.66±0.35 | 29.89±0.64 | 25.49 |
| Optimization-based | 26.02±0.36 | 31.64±1.75 | 27.93±0.47 | 34.82±0.96 | 30.10 |
| Self-sup.-selection | 22.36±0.30 | 28.56±0.50 | 25.35±0.27 | 32.57±0.13 | 27.21 |
| Moderate-DS | 27.24±0.36 | 32.90±0.31 | 29.06±0.28 | 35.89±0.53 | 31.27 |
| **GM Matching** | **27.96±1.60** | **35.76±0.82** | **34.11±0.65** | **40.91±0.84** | **34.69** |

Table 12: **Network Transfer** : A ResNet-50 proxy (pretrained on TinyImageNet) is used to find important samples from Tiny-ImageNet; which is then used to train a VGGNet-16 and ShuffleNet. We repeat the experiment across multiple corruption settings - clean; 20% Feature / Label Corruption and PGD attack when 20% and 30% samples are selected.

### 8.4 Additional Details on Baselines

Here, we detail the technical aspects of the baselines used in our experiments:

- **Random**: This approach involves randomly selecting a subset of the full dataset.
- **Herding** Welling (2009): This method selects data points that are closest to the class centers.
- **Forgetting** Toneva et al. (2018): Data points that are easily forgotten during optimization are chosen.
- **GraNd-score** Paul et al. (2021): Data points with larger loss gradient norms are included.
- **EL2N-score** Paul et al. (2021): This focuses on data points with larger norms of the error vector, which is the difference between the predicted class probabilities and the one-hot label encoding.
- **Optimization-based** Yang et al. (2022): This method uses the influence function Koh & Liang (2017) to select data points that minimize the generalization gap under strict constraints.
- **Self-sup.-selection** Sorscher et al. (2022): After self-supervised pre-training and clustering, data points are selected based on their distance to the nearest cluster centroid, with the number of clusters set to the number of classes to avoid tuning. Points with larger distances are chosen.

## 8.5 LEMMA 1 : VULNERABILITY OF IMPORTANCE SCORE BASED PRUNING

In the ideal setting, given a batch of i.i.d samples $\boldsymbol{\mu}_y = \boldsymbol{\mu}_y^{\mathcal{G}} = \mathbb{E}_{\mathbf{x} \sim \mathcal{D}_{\mathcal{G}}}(\mathbf{x})$. However, the presence of even a single grossly corrupted sample can cause the centroid estimate to deviate arbitrarily from the true mean. Consider a single grossly corrupt sample $(\mathbf{x}_i^{\mathcal{B}}, y_i)$ such that :

$$\mathbf{x}_i^{\mathcal{B}} = \sum_{(\mathbf{x}_i, y_i) \in \mathcal{D}} \mathbf{1}(y_i = y)\boldsymbol{\mu}_y^{\mathcal{B}} - \sum_{(\mathbf{x}_i, y_i) \in \mathcal{D} \backslash (\mathbf{x}_i^{\mathcal{B}}, y_i)} \mathbf{1}(y_i = y)\mathbf{x}_i \tag{11}$$

resulting in shifting the estimated centroid $\Delta\boldsymbol{\mu}_y = \boldsymbol{\mu}_y^{\mathcal{B}} - \boldsymbol{\mu}_y^{\mathcal{G}}$

**Lemma 1.** *A single gross corrupted sample* (11) *causes the importance scores to deviate arbitrarily:*

$$\Delta d(\mathbf{x}_i, y_i) = \|\Delta\boldsymbol{\mu}_y\|^2 - 2\left(\mathbf{x}_i - \boldsymbol{\mu}_y^{\mathcal{G}}\right)^T \Delta\boldsymbol{\mu}_y \tag{12}$$

*Implying, these methods yield the* **lowest possible asymptotic breakdown of 0**.

### 8.5.1 PROOF OF LEMMA 1

*Proof.* The original importance score without the corrupted sample is:

$$d(\mathbf{x}_i, y_i) = \|\mathbf{x}_i - \mu_y^{\mathcal{G}}\|_2^2 \tag{13}$$

The importance score with the corrupted sample affecting the centroid is:

$$d'(\mathbf{x}_i, y_i) = \|\mathbf{x}_i - \mu_y^{\mathcal{B}}\|_2^2 \tag{14}$$

We can calculate the deviation as:

$$\Delta d(\mathbf{x}_i, y_i) = d(\mathbf{x}_i, y_i) - d'(\mathbf{x}_i, y_i)$$
$$= \left(\mathbf{x}_i - \mu_y^{\mathcal{B}}\right)^T \left(\mathbf{x}_i - \mu_y^{\mathcal{B}}\right) - \left(\mathbf{x}_i - \mu_y^{\mathcal{G}}\right)^T \left(\mathbf{x}_i - \mu_y^{\mathcal{G}}\right)$$

The result follows by expanding and defining $\Delta\mu_y = \mu_y^{\mathcal{B}} - \mu_y^{\mathcal{G}}$ ∎

## 8.6 LEMMA 2: BOUNDING ESTIMATION ERROR FROM GM

In order to prove Theorem 1, we will first establish the following result which follows from the definition of GM; see also (Lopuhaa et al., 1991; Minsker et al., 2015; Cohen et al., 2016; Chen et al., 2017; Li et al., 2019; Wu et al., 2020; Acharya et al., 2022) for similar adaptations.

**Lemma 2.** *Given a set of $\alpha$-corrupted samples $\mathcal{D} = \mathcal{D}_\mathcal{G} \cup \mathcal{D}_\mathcal{B}$ ( Definition 1), and an $\epsilon$-approx. GM$(\cdot)$ oracle (4), then we have:*

$$\mathbb{E}\left\|\boldsymbol{\mu}^{\mathrm{GM}} - \boldsymbol{\mu}^{\mathcal{G}}\right\|^2 \leq \frac{8|\mathcal{D}_\mathcal{G}|}{(|\mathcal{D}_\mathcal{G}| - |\mathcal{D}_\mathcal{B}|)^2} \sum_{\mathbf{x}\in\mathcal{D}_\mathcal{G}} \mathbb{E}\left\|\mathbf{x} - \boldsymbol{\mu}^{\mathcal{G}}\right\|^2 + \frac{2\epsilon^2}{(|\mathcal{D}_\mathcal{G}| - |\mathcal{D}_\mathcal{B}|)^2} \tag{15}$$

*where, $\boldsymbol{\mu}^{\mathrm{GM}} = \mathrm{GM}(\{\mathbf{x}_i \in \mathcal{D}\})$ is the $\epsilon$-approximate GM over the entire ($\alpha$-corrupted) dataset; and $\boldsymbol{\mu}^{\mathcal{G}} = \frac{1}{|\mathcal{D}_\mathcal{G}|} \sum_{\mathbf{x}_i \in \mathcal{D}_\mathcal{G}} \mathbf{x}_i$ denotes the mean of the (underlying) uncorrupted set.*

### 8.6.1 PROOF OF LEMMA 2

*Proof.* Note that, by using triangle inequality, we can write:

$$\sum_{\mathbf{x}_i\in\mathcal{D}} \left\|\boldsymbol{\mu}^{\mathrm{GM}} - \mathbf{x}_i\right\| \geq \sum_{\mathbf{x}_i\in\mathcal{D}_\mathcal{B}} \left(\left\|\mathbf{x}_i\right\| - \left\|\boldsymbol{\mu}^{\mathrm{GM}}\right\|\right) + \sum_{\mathbf{x}_i\in\mathcal{D}_\mathcal{G}} \left(\left\|\boldsymbol{\mu}^{\mathrm{GM}}\right\| - \left\|\mathbf{x}_i\right\|\right)$$

$$= \left(\sum_{\mathbf{x}_i\in\mathcal{D}_\mathcal{G}} - \sum_{\mathbf{x}_i\in\mathcal{D}_\mathcal{B}}\right)\left\|\boldsymbol{\mu}^{\mathrm{GM}}\right\| + \sum_{\mathbf{x}_i\in\mathcal{D}_\mathcal{B}} \left\|\mathbf{x}_i\right\| - \sum_{\mathbf{x}_i\in\mathcal{D}_\mathcal{G}} \left\|\mathbf{x}_i\right\|$$

$$= \left(|\mathcal{D}_\mathcal{G}| - |\mathcal{D}_\mathcal{B}|\right)\left\|\boldsymbol{\mu}^{\mathrm{GM}}\right\| + \sum_{\mathbf{x}_i\in\mathcal{D}} \left\|\mathbf{x}_i\right\| - 2\sum_{\mathbf{x}_i\in\mathcal{D}_\mathcal{G}} \left\|\mathbf{x}_i\right\|. \tag{16}$$

Now, by definition (5); we have that:

$$\sum_{\mathbf{x}_i\in\mathcal{D}} \left\|\boldsymbol{\mu}^{\mathrm{GM}} - \mathbf{x}_i\right\| \leq \inf_{\mathbf{z}\in\mathcal{H}} \sum_{\mathbf{x}_i\in\mathcal{D}} \left\|\mathbf{z} - \mathbf{x}_i\right\| + \epsilon \leq \sum_{\mathbf{x}_i\in\mathcal{D}} \left\|\mathbf{x}_i\right\| + \epsilon \tag{17}$$

Combining these two inequalities, we get:

$$\left(|\mathcal{D}_\mathcal{G}| - |\mathcal{D}_\mathcal{B}|\right)\left\|\boldsymbol{\mu}^{\mathrm{GM}}\right\| \leq \sum_{\mathbf{x}_i\in\mathcal{D}} \left\|\mathbf{x}_i\right\| - \sum_{\mathbf{x}_i\in\mathcal{D}} \left\|\mathbf{x}_i\right\| + 2\sum_{\mathbf{x}_i\in\mathcal{D}_\mathcal{G}} \left\|\mathbf{x}_i\right\| + \epsilon \tag{18}$$

This implies:

$$\left\|\boldsymbol{\mu}^{\mathrm{GM}}\right\| \leq \frac{2}{\left(|\mathcal{D}_\mathcal{G}| - |\mathcal{D}_\mathcal{B}|\right)} \sum_{\mathbf{x}_i\in\mathcal{D}_\mathcal{G}} \left\|\mathbf{x}_i\right\| + \frac{\epsilon}{\left(|\mathcal{D}_\mathcal{G}| - |\mathcal{D}_\mathcal{B}|\right)} \tag{19}$$

Squaring both sides,

$$\left\|\boldsymbol{\mu}^{\mathrm{GM}}\right\|^2 \leq \left[\frac{2}{\left(|\mathcal{D}_\mathcal{G}| - |\mathcal{D}_\mathcal{B}|\right)} \sum_{\mathbf{x}_i\in\mathcal{D}_\mathcal{G}} \left\|\mathbf{x}_i\right\| + \frac{\epsilon}{\left(|\mathcal{D}_\mathcal{G}| - |\mathcal{D}_\mathcal{B}|\right)}\right]^2 \tag{20}$$

$$\leq 2\left[\frac{2}{\left(|\mathcal{D}_\mathcal{G}| - |\mathcal{D}_\mathcal{B}|\right)} \sum_{\mathbf{x}_i\in\mathcal{D}_\mathcal{G}} \left\|\mathbf{x}_i\right\|\right]^2 + 2\left[\frac{\epsilon}{\left(|\mathcal{D}_\mathcal{G}| - |\mathcal{D}_\mathcal{B}|\right)}\right]^2 \tag{21}$$

Where, the last step is a well-known consequence of triangle inequality and AM-GM inequality. Taking expectation on both sides, we have:

$$\mathbb{E}\left\|\boldsymbol{\mu}^{\mathrm{GM}}\right\|^2 \leq \frac{8}{\left(|\mathcal{D}_\mathcal{G}| - |\mathcal{D}_\mathcal{B}|\right)^2} \sum_{\mathbf{x}_i\in\mathcal{D}_\mathcal{G}} \mathbb{E}\left\|\mathbf{x}_i\right\|^2 + \frac{2\epsilon^2}{\left(|\mathcal{D}_\mathcal{G}| - |\mathcal{D}_\mathcal{B}|\right)^2} \tag{22}$$

Since, GM is **translation equivariant**, we can write:

$$\mathbb{E}\left[\text{GM}\left(\left\{\mathbf{x}_i - \boldsymbol{\mu}^{\mathcal{G}} | \mathbf{x}_i \in \mathcal{D}\right\}\right)\right] = \mathbb{E}\left[\text{GM}\left(\left\{\mathbf{x}_i | \mathbf{x}_i \in \mathcal{D}\right\}\right) - \boldsymbol{\mu}^{\mathcal{G}}\right] \tag{23}$$

Consequently we have that :

$$\mathbb{E}\left\|\boldsymbol{\mu}^{\text{GM}} - \boldsymbol{\mu}^{\mathcal{G}}\right\|^2 \leq \frac{8}{\left(|\mathcal{D}_{\mathcal{G}}| - |\mathcal{D}_{\mathcal{B}}|\right)^2} \sum_{\mathbf{x}_i \in \mathcal{D}_{\mathcal{G}}} \mathbb{E}\left\|\mathbf{x}_i - \boldsymbol{\mu}^{\mathcal{G}}\right\|^2 + \frac{2\epsilon^2}{\left(|\mathcal{D}_{\mathcal{G}}| - |\mathcal{D}_{\mathcal{B}}|\right)^2}$$

This concludes the proof. ∎

## 8.7 PROOF OF THEOREM 1

We restate the theorem for convenience:

**Theorem** 1 Suppose that, we are given, a set of $\alpha$-corrupted samples $\mathcal{D} = \mathcal{D}_\mathcal{G} \cup \mathcal{D}_\mathcal{B}$ (Definition 1) and an $\epsilon$ approx. $\text{GM}(\cdot)$ oracle (4). Further assume that $\|\mathbf{x}\| \leq R \ \forall \mathbf{x} \in \mathcal{D}$ for some constant $R$. Then, GM MATCHING guarantees that the mean of the selected $k$-subset $\mathcal{D}_\mathcal{S} \subseteq \mathcal{D}$ converges to a $\delta$-neighborhood of the uncorrupted (true) mean $\boldsymbol{\mu}^\mathcal{G} = \mathbb{E}_{\mathbf{x} \in \mathcal{D}_\mathcal{G}}(\mathbf{x})$ at the rate $\mathcal{O}(\frac{1}{k})$ such that:

$$\delta^2 = \mathbb{E}\left\|\frac{1}{k}\sum_{\mathbf{x}_i \in \mathcal{D}_\mathcal{S}}\mathbf{x}_i - \boldsymbol{\mu}^\mathcal{G}\right\|^2 \leq \frac{8|\mathcal{D}_\mathcal{G}|}{(|\mathcal{D}_\mathcal{G}| - |\mathcal{D}_\mathcal{B}|)^2}\sum_{\mathbf{x} \in \mathcal{D}_\mathcal{G}}\mathbb{E}\left\|\mathbf{x} - \boldsymbol{\mu}^\mathcal{G}\right\|^2 + \frac{2\epsilon^2}{(|\mathcal{D}_\mathcal{G}| - |\mathcal{D}_\mathcal{B}|)^2} \quad (24)$$

*Proof.* To prove this result, we first show that GM MATCHING converges to $\boldsymbol{\mu}_\epsilon^{\text{GM}}$ at $\mathcal{O}(\frac{1}{k})$. It suffices to show that the error $\delta = \left\|\boldsymbol{\mu}_\epsilon^{\text{GM}} - \frac{1}{k}\sum_{\mathbf{x}_i \in \mathcal{S}}\mathbf{x}_i\right\| \to 0$ asymptotically. We will follow the proof technique in (Chen et al., 2010) mutatis mutandis to prove this result. We also assume that $\mathcal{D}$ contains the support of the resulting noisy distribution.

We start by defining a GM-centered marginal polytope as the convex hull –

$$\mathcal{M}_\epsilon := \text{conv}\left\{\mathbf{x} - \boldsymbol{\mu}_\epsilon^{\text{GM}} \ |\mathbf{x} \in \mathcal{D}\right\} \quad (25)$$

Then, we can rewrite the update equation (8) as:

$$\boldsymbol{\theta}_{t+1} = \boldsymbol{\theta}_t + \boldsymbol{\mu}_\epsilon^{\text{GM}} - \mathbf{x}_{t+1} \quad (26)$$

$$= \boldsymbol{\theta}_t - (\mathbf{x}_{t+1} - \boldsymbol{\mu}_\epsilon^{\text{GM}}) \quad (27)$$

$$= \boldsymbol{\theta}_t - \left(\arg\max_{\mathbf{x} \in \mathcal{D}}\langle\boldsymbol{\theta}_t, \mathbf{x}\rangle - \boldsymbol{\mu}_\epsilon^{\text{GM}}\right) \quad (28)$$

$$= \boldsymbol{\theta}_t - \arg\max_{\mathbf{m} \in \mathcal{M}_\epsilon}\langle\boldsymbol{\theta}_t, \mathbf{m}\rangle \quad (29)$$

$$= \boldsymbol{\theta}_t - \mathbf{m}_t \quad (30)$$

Now, squaring both sides we get :

$$\|\boldsymbol{\theta}_{t+1}\|^2 = \|\theta_t\|^2 + \|\mathbf{m}_t\|^2 - 2\langle\boldsymbol{\theta}_t, \mathbf{m}_t\rangle \quad (31)$$

rearranging the terms we get:

$$\|\boldsymbol{\theta}_{t+1}\|^2 - \|\theta_t\|^2 = \|\mathbf{m}_t\|^2 - 2\langle\boldsymbol{\theta}_t, \mathbf{m}_t\rangle \quad (32)$$

$$= \|\mathbf{m}_t\|^2 - 2\|\mathbf{m}_t\|\|\boldsymbol{\theta}_t\|\langle\frac{\boldsymbol{\theta}_t}{\|\boldsymbol{\theta}_t\|}, \frac{\mathbf{m}_t}{\|\mathbf{m}_t\|}\rangle \quad (33)$$

$$= 2\|\mathbf{m}_t\|\left(\frac{1}{2}\|\mathbf{m}_t\| - \|\boldsymbol{\theta}_t\|\langle\frac{\boldsymbol{\theta}_t}{\|\boldsymbol{\theta}_t\|}, \frac{\mathbf{m}_t}{\|\mathbf{m}_t\|}\rangle\right) \quad (34)$$

Assume that $\|\mathbf{x}_i\| \leq r \ \forall \mathbf{x}_i \in \mathcal{D}$. , Then we note that,

$$\|\mathbf{x}_i - \boldsymbol{\mu}_\epsilon^{\text{GM}}\| \leq \|\mathbf{x}_i\| + \|\boldsymbol{\mu}_\epsilon^{\text{GM}}\| \leq 2r$$

Plugging this in, we get:

$$\|\boldsymbol{\theta}_{t+1}\|^2 - \|\theta_t\|^2 \leq 2\|\mathbf{m}_t\|\left(r - \|\boldsymbol{\theta}_t\|\langle\frac{\boldsymbol{\theta}_t}{\|\boldsymbol{\theta}_t\|}, \frac{\mathbf{m}_t}{\|\mathbf{m}_t\|}\rangle\right) \quad (35)$$

Recall that, $\boldsymbol{\mu}_\epsilon^{\text{GM}}$ is guaranteed to be in the relative interior of $\text{conv}\{\mathbf{x} \ |\mathbf{x} \in \mathcal{D}\}$ (Lopuhaa et al., 1991; Minsker et al., 2015). Consequently, $\exists \kappa$-ball around $\boldsymbol{\mu}_\epsilon^{\text{GM}}$ contained inside $\mathcal{M}$ and we have $\forall t > 0$

$$\langle\frac{\boldsymbol{\theta}_t}{\|\boldsymbol{\theta}_t\|}, \frac{\mathbf{m}_t}{\|\mathbf{m}_t\|}\rangle \geq \kappa > 0 \quad (36)$$

This implies, $\forall t > 0$

$$\|\boldsymbol{\theta}_t\| \leq \frac{r}{\kappa} \tag{37}$$

Expanding the value of $\boldsymbol{\theta}_t$ we have:

$$\left\|\boldsymbol{\theta}_k\right\| = \left\|\boldsymbol{\theta}_0 + k\boldsymbol{\mu}_\epsilon^{\text{GM}} - \sum_{i=1}^{k} \mathbf{x}_k\right\| \leq \frac{r}{\kappa} \tag{38}$$

Apply Cauchy Schwartz inequality:

$$\left\|k\boldsymbol{\mu}_\epsilon^{\text{GM}} - \sum_{i=1}^{k} \mathbf{x}_k\right\| \leq \left\|\boldsymbol{\theta}_0\right\| + \frac{r}{\kappa} \tag{39}$$

normalizing both sides by number of iterations $k$

$$\left\|\boldsymbol{\mu}_\epsilon^{\text{GM}} - \frac{1}{k}\sum_{i=1}^{k} \mathbf{x}_k\right\| \leq \frac{1}{k}\left(\left\|\boldsymbol{\theta}_0\right\| + \frac{r}{\kappa}\right) \tag{40}$$

Thus, we have that GM MATCHING converges to $\boldsymbol{\mu}_\epsilon^{\text{GM}}$ at the rate $\mathcal{O}(\frac{1}{k})$.

Combining this with Lemma 2, completes the proof.

$\blacksquare$

