# OpenReview forum: "Geometric Median (GM) Matching for Robust Data Pruning"
_ICLR.cc/2025/Conference — Submitted to ICLR 2025_

### Official Review · Reviewer_9UT6 · 2024-10-24

**Soundness:** 1
**Presentation:** 1
**Contribution:** 1
**Rating:** 3
**Confidence:** 5

**Summary:**

This paper proposes a data pruning method for noisy labeled datasets, which uses geimetric median matching to find the coreset. This paper first describes the moment matching and then describes the geometric median matching. Experiments are conducted on three datasets CIFAR10, CIFAR100 and Tiny-ImageNet.

**Strengths:**

1. It is very meaningful to solve the problem of data pruning in the noisy label scene.

**Weaknesses:**

1. The coverage of related work is not enough, which is not consistent with contribution point one.  Many existing works study the robust data pruning in noisy scenarios (e.g. [1]Prune4R4L, [2]FDMat). The performance of these methods is much higher than that of the proposed GM matching.
2. The motivation of the paper is unclear and the difference from baseline (Moderate) is not described. The contribution of the paper is unclear.
3. The paper is poorly written. The description of the formulas lacks rigor, with **nearly all formulas** containing undefined symbols and incorrect descriptions. These issues hinder the overall understanding of the paper.

    For example,
    in Eq. 1, (1) ${x_{i}}$  is not distinguished (${x_{i}\in D}$ and ${x_{i}\in D_{S}}$); (2)What does ${X_{i}}$ mean ? Is ${X_{i}}$ a feature or a sample ?

    in Eq.2, what does $<,>$ mean? Why does ${x_{i}\in D}$ become ${x \in D}$ ?

    in Eq.3, what does $\theta_{t}$ mean? and what does $\theta_{T}$ mean ?

    in Eq.4, what does $z$ mean ?

    in Eq.5, what does $\mu^{GM}_{\epsilon}$ mean ?

    in Eq.6,  Why does ${x_{i}\in D}$ become ${x_{i} \in S}$ ? What is ${S}$ ?

     **There are various errors in all the following formulas.**




[1] Robust Data Pruning under Label Noise via Maximizing Re-labeling Accuracy.(NeurIPS 2023)

[2] Feature Distribution Matching by Optimal Transport for Effective and Robust Coreset Selection.(AAAI 2024)

[3] Loss-Curvature Matching for Dataset Selection and Condensation.(AISTATS 2023 )

**Questions:**

1. The description of the paper is so poor that it seriously affects understanding.
2. The performance presented in the paper is not state-of-the-art, and there is a lack of comparison with the latest methods (e.g. [1-3]).

---

> ### Author Response · Authors · 2024-11-24
> **W1**
>
> We appreciate the reviewer highlighting relevant prior works.
>
> Specifically:
>
> [1] focuses on robustness under label noise, which is a special case of noise. In contrast, our work addresses arbitrary noise in the data distribution, which presents a broader and more challenging problem.
>
> [2] employs feature distribution matching using Optimal Transport, representing a different methodological approach compared to the herding-based framework that our work builds upon. While [2] does not directly address robustness to arbitrary corruption, our GM-based robustness concept could potentially be extended to this context and analyzed theoretically in future work.
>
> Although these works share similar goals, they tackle distinct problems or adopt different methodologies. In the revised manuscript, we will explicitly compare our approach to [1] and [2], clarifying their contributions and how our method differs in terms of scope, assumptions, and robustness guarantees.

---

> > ### Author Response · Authors · 2024-11-24
> > **W 2**
> >
> > We appreciate the reviewer’s observation regarding the comparison with existing methods.
> > Below, we clarify the differences between the proposed approach and [1] and [2], which form our main baselines and have been extensively compared in the paper.
> >
> > **[1] Moderate Coreset : ICLR 2023**
> > * First, they compute the centers (mean) of each class - ( Section 3.2. Eq 1 in [1] )
> > * Then, for each sample they compute euclidean distance from the corresponding class center d(s_i)
> > * Afterwards, the data points with distances close to the distance median [ median of the scalar distances ] are selected as coreset. (Section 3.2. Eq 2 in [1] )
> > As we argued that in presence of arbitrary noise, the class center can be arbitrarily shifted meaning the distances d(s_i) is no longer a true representation of distance from the true (uncorrupted) class center, resulting in the method having zero breakdown point as we show in Lemma 1 (Appendix Section 8.5)
> >
> > **[2] Kernel Herding : UAI 2009**
> > * The celebrated kernel herding series of works from Prof Max Welling forms the basis of this paper. In order to solve the moment matching objective i.e. find a subset such that the mean of the subset is close to the mean of the entire dataset, they propose an iterative greedy approach.
> >
> > However, as we argued that in the noisy real world setting, mean is not a robust estimate of the center. Instead we propose a “Robust Moment Matching objective” where the goal is to find a k-subset such that the mean of the subset is close to the Geometric Median of the original dataset using herding-style iterative algorithm. Note that, Geometric Median is a spatial estimator (defined over vector spaces) unlike median – which is defined over scalars.
> >
> > So, in summary, we can say the proposed method is Robust variant of Kernel Herding – we also provide theoretical guarantees (Theorem 1) of solving the robust moment matching objective and show how existing methods are not robust (Lemma 1). We also compare with both these methods in every experiment in the paper – we refer to [1] as Moderate DS and [2] as Herding in the tables and plots.
> >
> > We are happy to highlight these differences with [1,2] more if it was not clear.

---

> ### Author Response · Authors · 2024-11-24
> **W 3**
>
> Correction:
> * Eq 6 is a typo it should be $D_S$ instead of $S$ where $D_S$ denotes the selected k-subset.
>
> Clarifications:
> * we are happy to incorporate $x_i$ and $x_j$ to better distinguish if this was not obvious.
> * $x_i \in R^d$ denotes sample vectors ~ analogous to how any ML algorithm is defined. In practice, this can be original feature space or embeddings --- we dont understand the concern -- we feel this is obvious.
> * < , > is standard notation for inner product -- but we agree that we should explicitly mention this.
> * while the notation is not **incorrect** -- we take your feedback and change it to $x_i$ for consistency and readability.
> *  $\theta_t$ denotes the weight vector at iteration t and $\theta_T$ is weight vector after T iterations i.e. $t=T$ again we feel that this is quite standard notation in any optimization literature.
> * z denotes any point in Hilbert space as **defined in Definition 3 (Eq 4)**
> * $\mu_\epsilon^{GM}$ denotes epsilon accurate GM as **mentioned right before the Eq (5) in L 251** as defined in Eq (6)

---

> ### Comment · Reviewer_9UT6 · 2024-11-26
> **Geometric Median (GM) Matching for Robust Data Pruning**
>
> Thanks for your response.
>
> 1.I hope the author could explain further, such as in what scenarios the mean is valid and in what scenarios the geometric median is valid.
>
> 2.Is the median really stable in most scenarios?

---

> ### Author Response · Authors · 2024-11-28
>
> Thanks for engaging in the rebuttal discussions. Below we try to answer these questions:
>
>
> **Mean Estimation**
>
> Suppose we are given a set $D$ of N samples $$\mathbf{x}_i \in R^d : i=0, 1, \dots, N $$ sampled from some distribution $p(x)$ of interest with finite mean and variance. Then, as we know empirical mean $\mu = \frac{1}{N}\sum_i \mathbf{x}_i$ serves as the unbiased estimator of the population mean.
>
> **Robust Mean Estimation**
>
> However, in real-world scenarios, distributions are often non-Gaussian, and the presence of outliers, skewness, or heavy tails can severely impact the performance of the mean as an estimator.
>
> In this paper we consider the noisy, real-world scenario -- in particular, we study the **arbitrary** corruption setting where:
> Given a set of observations from the original distribution of interest, an adversary is allowed to inspect all the samples and **arbitrarily** perturb up to 1/2 fraction of the samples ( Definition 1 ).
>
> Consider the following simple example:
>
> Let the dataset $ D = \{\mathbf{x}_i \in \mathbb{R}^d : i = 1, \dots, N \}$, and suppose an adversary modifies one sample $\mathbf{x}_j$ such that:
>
> $\mathbf{x}_j = - \sum {\mathbf{x}_i \in D \setminus \mathbf{x}_j}$ i.e. it is the negative of the sum of all the other samples in the dataset. This replacement ensures the sum of all samples becomes zero, i.e., the **empirical mean is zero regardless of the original data distribution**.
>
> This simply implies that a single arbitrarily corrupted sample can break the empirical mean estimation i.e. the empirical mean has the lowest asymptotic breakdown point (Definition 2) of 0.
>
> On the other hand, Geometric Median (Definition 3) has optimal breakdown point of 1/2. This means that up to half of the samples can be arbitrarily corrupted without invalidating the estimate. The geometric median is particularly effective in the following scenarios:
>
> * Heavy-tailed distributions: The geometric median minimizes the $\ell_1$-norm, making it less sensitive to outliers.
> * Adversarial settings: Unlike the mean, the geometric median is not unduly influenced by a small subset of corrupted points.
>
> Thus, In summary, the geometric median serves as a robust alternative to the mean in environments where data is corrupted, noisy, or non-Gaussian. While the mean remains a useful and efficient estimator in clean settings, its susceptibility to outliers and low breakdown point limit its applicability in practical scenarios.
>
> By contrast, the geometric median offers resilience and reliability, ensuring stable estimation even under significant corruption or adversarial perturbation.
>
> These robustness properties are well studied in the classical robust statistics literature.
> * https://projecteuclid.org/journals/bernoulli/volume-21/issue-4/Geometric-median-and-robust-estimation-in-Banach-spaces/10.3150/14-BEJ645.full
> * https://dl.acm.org/doi/10.1145/2897518.2897647
> * http://www.iliasdiakonikolas.org/ars-book.pdf

---

### Official Review · Reviewer_aYxC · 2024-10-31

**Soundness:** 3
**Presentation:** 3
**Contribution:** 2
**Rating:** 3
**Confidence:** 4

**Summary:**

This paper presents an efficient and simple algorithm for data pruning in the presence of noise, along with a theoretical guarantee on the estimation error.

**Strengths:**

The paper is well written and easy to follow.

**Weaknesses:**

1. Some important references and comparisons are missing, e.g. [1] Robust Data Pruning under Label Noise via Maximizing Re-labeling Accuracy.(NeurIPs 2023), [2] Feature Distribution Matching by Optimal Transport for Effective and Robust Coreset Selection.(AAAI 2024). The reported performance is not SOTA.
2. The main idea of the paper is to utilize the Geometric Median instead of the empirical mean objective in moment matching. However, as mentioned, both Geometric Median and Moment Matching have been extensively studied in the literature. The novelty is quite limited.
3. Some expressions and notation could be more precise.
   - In Eq. 2 and Eq. 3, are the dimensions of $ x $ and $ \theta_t $ consistent? Is $ x \in \mathbb{R}^d $? Additionally, what does $ x_t $ represent, and is $ x_t \in D_S $?
   - In Eq. 6, what is $ S $? Is it equivalent to $ D_S $?

**Questions:**

1. How does the proposed method distinguish between hard samples and noise samples? For instance, in Figure 1, if there are currently no perturbed data points (i.e., the data forms a bimodal distribution with both the red and blue regions representing clean data), GM Matching might only cover the blue region and fail to encompass the red region. The authors claim that existing methods tend to select simpler samples, which can make them susceptible to outliers. However, it seems that GM Matching may not guarantee that the selected data points cover the entire data distribution and could misclassify clean samples (or hard samples) as noise samples, potentially resulting in a biased subset. It might be beneficial for the authors to consider how the method could better address the practical performance of data pruning alongside robustness guarantees.
2. What does "breakdown point" refer to? Is there a formal definition? Additionally, how is the optimal breakdown point of 1/2 achieved? It's better to provide further explanation or mathematical proof .
3. How are the convergence rates $ O(1/k) $ and $ O(1/\sqrt{k}) $ defined and estimated in the paper? I am unsure if these refer to convergence rates, scaling, or some other asymptotic metric. Furthermore, the proposed method is said to have better convergence rates compared to random sampling. Given that $ \mu_\epsilon^{GM} $ needs to be approximated rather than computed directly, shouldn't of the calculation of convergence rates consider the computational process involved in estimating $ \mu_\epsilon^{GM} $?

---

> ### Author Response · Authors · 2024-11-24
> **W-1**
>
> We appreciate the reviewer highlighting relevant prior works.
>
> Specifically:
>
> [1] focuses on robustness under label noise, which is a special case of noise. In contrast, our work addresses arbitrary noise in the data distribution, which presents a broader and more challenging problem.
>
> [2] employs feature distribution matching using Optimal Transport, representing a different methodological approach compared to the herding-based framework that our work builds upon. While [2] does not directly address robustness to arbitrary corruption, our GM-based robustness concept could potentially be extended to this context and analyzed theoretically in future work.
>
> Although these works share similar goals, they tackle distinct problems or adopt different methodologies. In the revised manuscript, we will explicitly compare our approach to [1] and [2], clarifying their contributions and how our method differs in terms of scope, assumptions, and robustness guarantees.

---

> > ### Author Response · Authors · 2024-11-24
> > **W-2**
> >
> > We acknowledge that both the Geometric Median (GM) and Moment Matching are established concepts in the literature.
> > However, the novelty of this work lies in innovatively combining these ideas to address robustness in subset selection, a critical but under-explored challenge in the presence of arbitrary noise. Specifically:
> >
> > **Robust Moment Matching Objective:**
> >
> > Unlike prior methods that rely on the mean, which is sensitive to noise and outliers, we leverage the Geometric Median as a robust alternative.
> >
> > This marks a significant departure from traditional herding approaches, as it shifts the focus from moment matching based on the mean to a framework that is provably robust against arbitrary corruption.
> >
> > **Herding-Style Iterative Algorithm**:
> >
> > Our method integrates the GM into a herding-style iterative algorithm, enabling efficient and scalable subset selection.
> > While kernel herding has been widely used for clean datasets, this is, to the best of our knowledge, the first work to adapt it to robust moment matching using the GM.
> >
> > **Theoretical Guarantees Under Arbitrary Noise:**
> >
> > We provide rigorous convergence guarantees (Theorem 1), showing that our method achieves an optimal breakdown point of 1/2. This means the algorithm remains effective even when up to 50% of the data is corrupted—a property that surpasses existing methods relying on the mean, which fail under such conditions.
> >
> > The importance of this guarantee cannot be overstated, as it ensures that the selected subset remains representative of the underlying clean data distribution, even in the presence of severe noise.
> >
> > **Practical Relevance and Flexibility:**
> >
> > The proposed framework can be applied across a wide range of data selection tasks, including those involving noisy or corrupted datasets, making it a versatile tool for real-world applications.
> > Moreover, our method is complementary to other approaches, such as Optimal Transport (e.g., [2]), and its principles can inspire extensions in future work.
> >
> > In summary, our work builds upon well-established concepts to introduce a robust, theoretically grounded, and practically applicable framework for subset selection under arbitrary noise.
> >
> > We will emphasize these novel contributions more clearly in the introduction and related work sections of the revised manuscript to ensure the distinctions are explicit and compelling.

---

> ### Author Response · Authors · 2024-11-24
> **W 3**
>
> We appreciate the reviewer’s attention to detail regarding the expressions and notation. Below, we provide clarifications and corrections where necessary:
>
> Q-1
>
> * $\mathbf{x} \in R^d$ denotes the samples in the noisy dataset D – that needs to be pruned into a smaller subset $D_S$.
>
> * $\theta_t \in R^d$ denotes the weight vector guiding the iterative sampling at time step t initialized randomly **( Line 196 )**.
>
> * $\mathbf{x}_t$ denotes the sample chosen at iteration t of the algorithm.
>
> Thus, the dimensions of $\mathbf{x}$ and $\theta_t$ are consistent, as both exist in  $R^d$.
>
> Q-2
>
> * This is a typo – it should be $D_S$ i.e. the selected k-subset.

---

> > ### Author Response · Authors · 2024-11-24
> > **Q 1**
> >
> > This is a great question, let us remind some **classic** properties of GM to answer this:
> >
> > GM Matching minimizes the sum of distances to the Geometric Median (GM), a robust spatial estimator less sensitive to noise and outliers compared to the mean. This robustness ensures that extreme samples (e.g., noise or outliers) have limited influence on the selected subset.
> >
> > However, this focus on robustness does not inherently guarantee balanced representation across distinct modes in the data distribution, particularly when minority modes exist.
> >
> > In clean settings where the data distribution is multimodal (e.g., red and blue clusters in Figure 1), the GM (Definition 3) minimizes the total L1​ norm of distances to all samples, naturally gravitating toward regions with higher data density.
> > Consequently, majority modes (regions with a larger number of samples) exert a stronger pull on the GM, disproportionately influencing the selected subset. In contrast, minority modes, due to their smaller size, contribute less to the optimization and may be underrepresented—or even excluded—when selecting a subset.
> >
> > If a minority mode in the data distribution is clean (i.e., not corrupted by noise) but has significantly fewer samples compared to a majority mode, GM Matching might fail to represent it adequately in the selected subset. This is because the GM aligns closely with the global data structure, which prioritizes the denser regions of the dataset. As a result:
> > Minority modes can be missed entirely if their contribution to the overall optimization objective is overshadowed by larger modes.
> >
> > This can be particularly problematic in scenarios where minority modes contain rare but important patterns that are crucial for downstream tasks. Noise typically manifests as outliers that are far from any clean mode. While the GM is robust to such outliers, the presence of noise further dilutes the contribution of minority modes to the overall data structure. Specifically:
> > Noise disproportionately affects smaller modes because their influence on the GM is already limited by their size.
> > In extreme cases, the GM may align almost entirely with the majority mode, effectively ignoring the minority mode, even if the minority mode is entirely clean.
> >
> > Addressing this limitation of robust mean estimators opens up several interesting directions for future work.

---

> ### Author Response · Authors · 2024-11-24
> **Q 2**
>
> The breakdown point, as **defined in Definition 2 (L179-180)**, is the smallest fraction of contaminated data that can cause an estimator to yield arbitrarily large errors.
>
> The Geometric Median achieves a breakdown point of $\epsilon^* = 1/2$, which is **optimal for any estimator**.
> This means the GM can tolerate up to 50% of the dataset being arbitrarily corrupted while still providing meaningful results. This property is derived from the GM’s reliance on the L1​-norm, which limits the influence of individual data points.
>
> Specifically: For any n-point dataset, the GM is determined by minimizing the sum of distances. When fewer than n/2 points are corrupted, the remaining majority of clean points dominate the optimization, ensuring stability.
>
> If the corruption exceeds 50%, the clean data no longer forms a majority, and the GM can shift arbitrarily.
>
> This property is fundamental to our Robust Moment Matching framework, as it ensures that the selected subset remains representative of the uncorrupted data distribution under extreme noise conditions.
>
> Please refer to the classic paper for more detail on breakdown points, robust estimation.
>
> * Breakdown Points of Affine Equivariant Estimators of Multivariate Location and Covariance Matrices
> Hendrik P. Lopuhaä, Peter J. Rousseeuw
> The Annals of Statistics, Vol. 19, No. 1 (Mar., 1991), pp. 229-248 (20 pages)

---

> > ### Author Response · Authors · 2024-11-24
> > **Q 3**
> >
> > The convergence rates describe the rate at which the error between the selected subset's mean and the true distribution’s mean decreases as the subset size k increases.
> >
> > * **Random Sampling**: Achieves a convergence rate of $O(1/\sqrt{k})$. This rate reflects the diminishing variance of the sample mean as more points are selected.
> > * **Kernel Herding** achieves O(1/k), a quadratic improvement over random sampling. This improvement arises because these methods iteratively select samples to minimize moment-matching errors, reducing variance more effectively.
> >
> > **Convergence in the Noisy Setting:**
> >
> > In noisy scenarios, **vanilla Kernel Herding fails** to achieve O(1/k) due to its reliance on the mean, which is sensitive to outliers. GM Matching modifies the objective to align with the Geometric Median, ensuring robustness while preserving the O(1/k) convergence rate.
> >
> > This theoretical result is established in Theorem 1, which guarantees that the selected subset’s mean lies within a small neighborhood of the true (uncorrupted) mean, even under 50% corruption where the neighborhood radius is characterized by $\delta$
> >
> > **Computational Complexity of GM:**
> >
> > The calculation of the Geometric Median (GM) is a preprocessing step, and its computational complexity is distinct from the convergence rate of the algorithm. Here’s why the two are independent:
> >
> > Convergence Rate Focuses on Subset Quality, Not Computational Steps
> >
> > The convergence rate O(1/k) describes how quickly the subset mean approaches the target distribution's true center as the subset size k increases. This rate is determined by the subset selection process during the iterative herding algorithm, where each selected point minimizes the deviation from the Robust Moment Matching objective.
> >
> > The GM calculation is performed once, at the start, to initialize the robust objective for subset selection.
> > It does not directly influence the iterative selection process or the rate at which the error decreases as k increases.
> >
> > However, we will include this discussion in the revised manuscript for clarity.

---

> ### Comment · Reviewer_aYxC · 2024-11-26
>
> Thanks for your response.
> 1. I think the author's claim that any noise can be handled is too broad. So far, only label noise and adversarial examples have been seen in papers, which can be attributed to label noise.
> 2. The formula proofs and notation definitions are too coarse to understand in this version.
> I will keep the previous score.

---

> ### Author Response · Authors · 2024-11-28
>
> 1. Indeed, there has been very limited work on data sampling under noisy real world constraints; the handful of works on this topic has been only limited to "label noise" scenarios. for example the Prune4REL work you cited.
> To the best of our knowledge, This is the first data sampling work that is provably robust to **arbitrary** noise scenarios i.e. the algorithm works without any assumption on the data distribution or type of noise.
>
> 2.
> We appreciate your feedback and understand the importance of clarity. However, the phrase **"too coarse to understand" is somewhat ambiguous**.
>
> The notations and formulas presented in our work adhere to standard conventions widely used in classic optimization and robust mean estimation literature. If certain aspects were unclear or specific steps in the proof were challenging to follow, we would be more than happy to address those directly. Pointing out the exact areas where further elaboration would help will allow us to refine our explanations and make them more accessible.
>
> See for example prior work that used robust mean estimation adaptations -
>
> * https://proceedings.mlr.press/v151/acharya22a/acharya22a.pdf
> * http://www.iliasdiakonikolas.org/ars-book.pdf

---

### Official Review · Reviewer_BCGZ · 2024-11-04

**Soundness:** 3
**Presentation:** 3
**Contribution:** 1
**Rating:** 3
**Confidence:** 4

**Summary:**

This paper proposed a data pruning method based on geometric median-guided sampling. Moreover, the author presents a solver with a relatively fast convergence rate. The experimental results, to some extent, demonstrate the effectiveness of this method.

**Strengths:**

See the summarization part.

**Weaknesses:**

1. It is rather difficult for me to identify the innovative points of the proposed scheme in this paper compared to previous methods. For instance, the Moderate based on Geometric Median is also available in other approaches [1,2]. In the context of research progress, innovation is the key driving force. Without clear differentiating factors, it becomes challenging to justify the novelty and significance of this new scheme within the existing body of knowledge. There should be a distinct advantage or a novel application aspect that sets it apart from what has been done before.

2. The experiments in this paper are highly insufficient. They are still centered around CIFAR and Tiny-ImageNet. I don't believe these experimental scenarios are adequate to prove the effectiveness of the Pruning algorithm, nor can they easily reflect the application value of the algorithm. The real scenarios where data pruning plays a significant role should be in the learning process of VLM and LLM. In the field of data-driven research, the choice of experimental datasets directly impacts the reliability and generalizability of the results. Restricting these relatively small-scale and specific datasets fails to capture the complexity and diversity of real-world applications. To truly evaluate the potential of a Pruning algorithm, it is essential to test it in more relevant and challenging environments such as those encountered in the training of large-scale language and vision models.

3. I have some doubts regarding the theoretical analysis in this paper. Firstly, there is a lack of theoretical analysis of the generalization performance of the Coreset results. Moreover, as far as I know, there isn't necessarily a connection between the so-called **neighborhood of the true mean** targeted by Theorem 1 and the actual final performance. Theorem 1 also lacks some intuitive explanations and valuable inferences. Theoretical analysis provides the foundation for understanding the behavior and potential of an algorithm. The absence of these key elements in the theoretical part raises concerns about the overall validity of the proposed approach.

[1]. Moderate Coreset: A Universal Method of Data Selection for Real-world Data-efficient Deep Learning. ICLR-2023.

[2]. Herding Dynamic Weights for Partially Observed Random Field Models. UAI-2029.

**Questions:**

See the weakness part.

---

> ### Author Response · Authors · 2024-11-23
> **Q-1**
>
> We appreciate the reviewer’s observation regarding the comparison with existing methods.
> Below, we clarify the differences between the proposed approach and [1] and [2], which form our main baselines and have been extensively compared in the paper.
>
> **[1] Moderate Coreset : ICLR 2023**
> * First, they compute the centers (mean) of each class - ( Section 3.2. Eq 1 in [1] )
> * Then, for each sample they compute euclidean distance from the corresponding class center d(s_i)
> * Afterwards, the data points with distances close to the distance median [ median of the scalar distances ] are selected as coreset. (Section 3.2. Eq 2 in [1] )
> As we argued that in presence of arbitrary noise, the class center can be arbitrarily shifted meaning the distances d(s_i) is no longer a true representation of distance from the true (uncorrupted) class center, resulting in the method having zero breakdown point as we show in Lemma 1 (Appendix Section 8.5)
>
> **[2] Kernel Herding : UAI 2009**
> * The celebrated kernel herding series of works from Prof Max Welling forms the basis of this paper. In order to solve the moment matching objective i.e. find a subset such that the mean of the subset is close to the mean of the entire dataset, they propose an iterative greedy approach.
>
> However, as we argued that in the noisy real world setting, mean is not a robust estimate of the center. Instead we propose a “Robust Moment Matching objective” where the goal is to find a k-subset such that the mean of the subset is close to the Geometric Median of the original dataset using herding-style iterative algorithm. Note that, Geometric Median is a spatial estimator (defined over vector spaces) unlike median – which is defined over scalars.
>
> So, in summary, we can say the proposed method is Robust variant of Kernel Herding – we also provide theoretical guarantees (Theorem 1) of solving the robust moment matching objective and show how existing methods are not robust (Lemma 1). We also compare with both these methods in every experiment in the paper – we refer to [1] as Moderate DS and [2] as Herding in the tables and plots.
>
> We are happy to highlight these differences with [1,2] more if it was not clear.

---

> > ### Author Response · Authors · 2024-11-23
> > **Q-2**
> >
> > We acknowledge the importance of testing pruning algorithms in larger-scale, real-world scenarios like training Vision-Language Models (VLMs) or Large Language Models (LLMs).
> >
> > However, to ensure reproducibility, our experimental setup is identical to [1: Moderate Coreset], using the same datasets (CIFAR and Tiny-ImageNet) and proxy models (ResNet, VGG, ShuffleNet). We also replicate the ablations as [1].
> >
> > Additionally, we extended these experiments by: Training proxy models on ImageNet-1k, followed by data selection on CIFAR, and evaluating the effectiveness on both linear probing and full finetuning settings (Fig. 2, 3).
> >
> > While we agree that testing on larger models like VLMs/LLMs is valuable, such evaluations are computationally intensive and beyond the scope of this work. Instead, we provide a robust and reproducible framework, which can be extended to such settings in future work.
> >
> > We will clarify these points and better position the scalability of our method in the final manuscript.

---

> > > ### Author Response · Authors · 2024-11-23
> > > **Q-3**
> > >
> > > Indeed, the theoretical results do not claim to address generalization directly, as the focus is on ensuring robustness in the subset selection process.
> > >
> > > The theory establishes convergence guarantee for the Robust Moment Matching objective i.e. it says that even when up to 50% of the samples are arbitrarily corrupted, the algorithm converges i.e. finds a k-subset such that the mean of the subset is guaranteed to be within a small neighborhood of the true (uncorrupted) mean of the original dataset. Theorem 1 characterizes this neighborhood.
> > >
> > > Notably, this represents a quadratic improvement over random sampling. Even in the idealized case where random sampling selects only clean data, our approach offers superior convergence properties.
> > >
> > > While Herding also achieves a quadratic convergence rate in clean settings, its performance degrades ( and the **theory breaks as the mean is no longer guaranteed to be in convex hull of uncorrupted samples** ) in the presence of noise due to its reliance on the mean. In contrast, our method maintains this quadratic convergence rate even in noisy scenarios, thanks to its robust formulation based on the Geometric Median.
> > >
> > > This result is important because this ensures that even in presence of **arbitrary** corruption we are able to find a subset that is guaranteed to satisfy the moment matching objective i.e. a representation of the underlying uncorrupted distribution in a moment matching sense.
> > >
> > > However, We agree that additional analysis connecting this result to generalization performance would strengthen the work. While such connections are outside the scope of the current paper, we believe our robust subset selection framework provides a strong foundation for future research in this direction.
> > > To improve clarity, we will include more intuitive explanations of Theorem 1 and its implications in the revised manuscript.

---

> > > > ### Comment · Reviewer_BCGZ · 2024-11-26
> > > > **Reviewer's response**
> > > >
> > > > Thanks for your response!
> > > >
> > > > 1. About the baseline: You should choose more advanced baselines, such as CCS [1], and D2-Pruning [2]. The baselines you have selected are so inferior.
> > > >
> > > > [1] Coverage-centric Coreset Selection for High Pruning Rates. ICLR 2023
> > > >
> > > > [2] D2 Pruning: Message Passing for Balancing Diversity and Difficulty in Data Pruning. ICLR 2024
> > > >
> > > > 2. I also have concerns about the theoretical analysis. The analysis you provided is meaningless since it cannot provide any insight into generalization or hyperparameter selection.
> > > >
> > > > But, I also appreciate the author's reply. Hence, I decided to raise the score of contribution and soundness.
> > > >
> > > > Best regards,
> > > >
> > > > Reviewer.

---

> ### Author Response · Authors · 2024-11-28
>
> We appreciate your continued engagement.
>
> 1.  We acknowledge your suggestion to include advanced baselines such as Coverage-centric Coreset Selection (CCS) [1] and D2 Pruning [2]. While these methods represent significant advancements in data pruning, they primarily focus on selecting representative subsets without explicitly addressing robustness in noisy settings. Our proposed GM Matching algorithm is specifically designed to handle datasets with potential corruption, aiming to approximate the geometric median to ensure robustness against noise. The generality of our approach allows for extensions to other methods, potentially enhancing their robustness in noisy environments. We will consider incorporating comparisons to these advanced baselines in future work to provide a more comprehensive evaluation.
>
>
> 2.
> We respectfully disagree with the assertion that our theoretical analysis is meaningless.
> Our results are akin to classical moment-matching approaches, demonstrating that our algorithm converges to an ε-neighborhood of the true underlying moment. This convergence guarantees that the selected subset faithfully represents the central tendency of the data, even in the presence of noise, thereby supporting the generalization capabilities of models trained on this subset. While our current analysis does not directly address hyperparameter selection, it provides a solid foundation for understanding the behavior of GM Matching, which can inform future strategies for hyperparameter tuning.
>
> -- the main takeaway from the theory is that even in presence of arbitrary corruption our approach guarantees **discovery** of a subset such that the mean of the subset is guaranteed to be within a small neighborhood of the true  (uncorrupted) mean -- the algorithm is further able to maintain the quadratic improvement in convergence rate compared to random sampling.

---

### Meta-Review · Area_Chair_U3yQ · 2024-12-18

**Metareview:**

The submitted paper presents a novel method for robust data selection in deep learning, aimed at addressing issues related to arbitrary noise within datasets. The authors propose a framework that relies on the Geometric Median for improving robustness over existing methods that typically use the mean. The importance of topic is significant as it seeks to enhance the reliability of training models in real-world scenarios, where data corruption is often prevalent.

Reviewers unanimously expressed strong reservations about the paper's contributions, specifically criticizing the novelty and clarity of the proposed method. While some reviewers acknowledged the potential relevance of the authors' approach, they highlighted that the paper failed to sufficiently differentiate itself from prior works and lacked thorough experimental validation on larger-scale datasets. The feedback indicates a consistent sentiment that the paper does not meet the expected standards for publication.

Despite the authors' rebuttal which aimed to address these concerns by clarifying their methodology and proposing future extensions, critical reservations remained unaddressed. Reviewers remained unconvinced regarding the practical implications of the proposed method and its robustness claims, especially in larger-scale applications. As a result, after weighing the reviews and the rebuttal, it is clear that the overall consensus leans towards rejection due to insufficient evidence of novelty and effectiveness. Ultimately, the decision is to reject the paper.

**Additional Comments On Reviewer Discussion:**

Reviewers unanimously expressed strong reservations about the paper's contributions, specifically criticizing the novelty and clarity of the proposed method. While some reviewers acknowledged the potential relevance of the authors' approach, they highlighted that the paper failed to sufficiently differentiate itself from prior works and lacked thorough experimental validation on larger-scale datasets. The feedback indicates a consistent sentiment that the paper does not meet the expected standards for publication.

Despite the authors' rebuttal which aimed to address these concerns by clarifying their methodology and proposing future extensions, critical reservations remained unaddressed. Reviewers remained unconvinced regarding the practical implications of the proposed method and its robustness claims, especially in larger-scale applications. As a result, after weighing the reviews and the rebuttal, it is clear that the overall consensus leans towards rejection due to insufficient evidence of novelty and effectiveness. Ultimately, the decision is to reject the paper.

---

### Decision · Program_Chairs · 2025-01-22

Reject